# Developmental low-dose bisphenol A exposure leads to extensive transcriptome female masculinization and male feminization later in life

Thomas Lind [1] ✉, Linda Dunder[2], Margareta H. Lejonklou[2], P. Monica Lind[2], Håkan Melhus[1] & Lars Lind[3]

## Abstract

**Background** Bisphenol A (BPA) is an endocrine disruptor, and exposure to low doses in utero has been associated with the development of metabolic diseases. Previous studies have suggested that bone marrow (BM) may be particularly susceptible to BPA exposure.
**Methods** Here, we investigate how developmental exposure to low levels of BPA affects the BM transcriptome and the blood metabolic profile in Fischer 344 rats later in life. We compare these effects to those observed in human metabolic syndrome (MetS) using a population-based cohort.
**Results** The results show an unexpectedly extensive sex-biased effect on the BM transcriptome from a BPA dose approximately eight times lower than the recent temporary European Food Safety Authority (EFSA) human tolerable daily intake (TDI) and a higher dose considered safe in 2015. BPA exposure induces sex-specific changes in gene expression, progressing toward a hypometabolic cancer-like state in females and a hypermetabolic autoimmunity-like state in males, with a blood metabolic profile that significantly overlaps with human MetS in a cross-sectional study.
**Conclusions** We conclude that developmental low-dose BPA exposure might induce metabolic syndrome specifically in males, possibly by affecting T cell activity in a sex-specific manner. Our study provides biologically plausible and convincing evidence for significant effects from low-dose BPA exposure, supporting the substantial lowering of the human BPA TDI by EFSA based on its critical effects on T cells.

## Plain Language Summary

Bisphenol A (BPA) is a chemical used to produce plastics, often used for storing food and beverages. Many studies have demonstrated a connection between BPA exposure and adverse health effects. Animal research indicates that low doses of BPA may have stronger impacts than higher doses. We gave pregnant rats drinking water with BPA levels similar to those considered safe for humans in 2015. We observed lasting effects in their offspring exposed to low-dose BPA. BPA caused notable female masculinization and male feminization. These sex-specific effects resembled those seen in people with metabolic syndrome, a set of health problems that raise the risk of diabetes and heart disease. Our results imply that reducing BPA use in food and beverage containers could help prevent these outcomes.

Bisphenol A (BPA) is a high-volume organic synthetic compound and an endocrine-disrupting chemical (EDC) with known effects on the action of estrogen, thyroid hormones, retinoids, and androgens. Various consumer products contain BPA in their production, which can leach and increase the risk of exposure. Therefore, it is unsurprising that BPA is present in the urine of a large portion of the human population, indicating chronic low-level exposure[1,2]. A recent overview of published meta-analyses suggests that BPA exposure has seriously affected human health, most convincingly preterm birth, obesity, allergy, and kidney disease[3]. Notably, BPA can cross the placental barrier and is present in cord blood, fetal tissue, placenta, and amniotic fluid, indicating fetal exposures occur[4,5]. Early-life exposure can influence disease outcomes throughout the entire lifespan of an organism, and the risk of developing diabetes, cardiovascular disorders, or cancer increases with age[6]. It is becoming increasingly apparent that low-dose developmental exposure to BPA can contribute to effects observed later in life[7]. Results from a collaborative project, called Consortium Linking Academic and Regulatory Insights on Toxicity of BPA (CLARITY–BPA), have been released (https://ntp.niehs.nih.gov/go/bpa). An important conclusion from this huge project, launched by three US federal agencies: the Food and Drug Administration (FDA), the National Institute of Environmental

[1]Department of Medical Sciences, Section of Clinical Pharmacogenomics and Osteoporosis, Uppsala University, Uppsala, Sweden. [2]Department of Medical Sciences, Occupational and Environmental Medicine, Uppsala University, Uppsala, Sweden. [3]Department of Public Health and Caring Sciences, Clinical Nutrition and Metabolism, Uppsala University, Uppsala, Sweden. ✉e-mail: thomas.lind@medsci.uu.se

Health Sciences (NIEHS), and the National Toxicology Program (NTP) together with 14 investigators at US institutions, was that the lower doses of BPA used (2.5–250 μg/kg body weight (BW)/day) elicited a more significant number of biological effects compared to the higher doses (2500–25,000 μg/kg BW/day)[8]. CLARITY-BPA demonstrated that developmental BPA exposure in Sprague–Dawley rats, within the range of estimated human daily exposure (0.01–5 μg/kg BW/day), affected brain and behavior, heart, ovary, mammary gland, prostate, liver, kidney, and the immune system in a sex-specific manner.

BPA is best known as an estrogen receptor agonist, and estrogen is the primary regulator of bone metabolism in both men and women[9]. This warrants studies on how BPA affects bone, mainly as this tissue was not investigated in the CLARITY-BPA program. A similar study of developmental BPA exposure in Wistar rats was conducted at the National Food Institute of Denmark; their lowest dose (25 μg/kg BW/day) showed signs of feminization (larger mammary gland epithelial trees) of males and masculinized neurobehaviours of females[10,11]. In addition, the lowest dose was the only one that affected both male and female long bones, with increased cortical thickness in males and increased bone length in females[12]. In the young siblings of the present study, BPA at a level comparable to human daily exposure induced a bone phenotype (thinner and shorter) in males, similar to that of diethylstilbestrol, but not to exogenous estrogen[13]. Notably, the bone phenotype in males disappeared with age, whereas the bones of females developed reduced stiffness[14]. Furthermore, these female rats also exhibited bone marrow fibrosis and chronic inflammation, even at a dose approximately eight times lower than the temporary EFSA human tolerable daily intake of 4 μg/kg body weight per day[15], suggesting that bone tissue may be particularly sensitive to low doses of BPA during development.

Bone maintains skeletal mass and regulates hematopoiesis, but it is also actively involved in whole-body metabolism, consuming large amounts of energy[16,17]. Skeletal glucose uptake equals or exceeds that of metabolic organs, such as the liver, muscle, and white adipose tissue[18,19]. The skeleton avidly takes up lipids from the circulation and is second only to the liver in this activity[20,21]. Bone marrow metabolism is impaired in insulin resistance and improves following physical activity[22]. Bone marrow is a vital organ, and its hematopoietic stem cell niche uniquely gives rise to various mature blood cell types and tissues. The bone marrow and spleen also serve as significant reservoirs of immune cells, maintaining immune homeostasis in peripheral tissues. In contrast to studies on peripheral blood, there is a knowledge gap regarding gene expression in adult murine hematopoietic organs[23].

In the present study, we exposed pregnant rats to drinking water containing a dose similar to daily human exposure (0.5 μg/kg BW/day) and a higher dose considered safe in 2015 (50 μg/kg BW/day)[15]. Female and male rat offspring were evaluated more than 48 weeks after their last exposure to BPA via their mother's milk. The primary objective of the present study was to investigate how developmental low-dose BPA exposure affects the rat bone marrow transcriptome and blood metabolome/lipidome later in life, and to compare this with humans who have metabolic syndrome (MetS) using a cross-sectional design. Bone marrow contains many cell types (adipocytes, immune cells, and stem cells) and is resistant to rapid transcriptome changes induced by dietary variations[24] or thermal adaption[25]. As BPA exposure has been linked to masculinization and feminization, an important focus was to compare how BPA affects the molecular differences that separate females from males. The work builds on previous analyses of the bone and metabolic phenotype of these adult rats and their young siblings[13,14,26–29].

Our results show that developmental exposure to low BPA doses has long-term, sex-specific effects on the rat metabolism. We further show that these effects overlap with those of humans diagnosed with MetS. Notably, BPA induces extensive female masculinization and male feminization of the bone marrow transcriptome.

## Methods
### Animals and housing
This work (C26/13) was approved by the Uppsala Ethical Committee on Animal Research, in accordance with Swedish legislation on animal experimentation (Animal Welfare Act, SFS 1998:56) and European Union legislation (Convention ETS 123 and Directive 2010/63/EU). The animals were treated humanely and with consideration for their well-being and comfort. Female Fischer 344/DuCrl (F344) rats, $n = 45$ (8–9-weeks-old), were mated simultaneously at Charles River, Germany, for housing at Uppsala University's animal facility. On gestational day (GD) 3.5, upon arrival at the laboratory, the dams were weighed, chip-marked, and dosing started immediately. The study was conducted in seven blocks, each separated by one week, with dose groups evenly distributed. Dams within each block were randomly assigned (simple randomization) to three dosing groups: 0 (CTRL; $n = 18$), 0.5 (BPA0.5; $n = 12$), or 50 (BPA50; $n = 15$) μg BPA/kg body weight per day, and housed individually. To monitor water intake, dams were housed separately. Polysulfone cages (Eurostandard IV) with glass water bottles were used to minimize background BPA exposure. Animals were housed in a temperature- and humidity-controlled room ($22 \pm 1$ °C and $55 \pm 5\%$ relative humidity), with a 12-h light/dark cycle and an air exchange rate of ten times per hour. Litters were adjusted to six pups per dam (three males and three females) on postnatal day (PND) 4. On the weaning day (PND 22), two males and two females were randomly selected (simple randomization) and placed in cages containing three offspring, separated by sex and dose. To prevent litter effects, pups of the same sex and dose came from different mothers. Food and water were provided ad libitum. Dams were fed a standard pelleted diet, RM3 (NOVA, SCB, Sollentuna, Sweden), while the offspring received RM1 (NOVA, SCB, Sollentuna, Sweden). The nutrient and phytoestrogen content of the feed batches used for dams and pups was specified by the manufacturer [RME3, batch 9987: 11.2 and <10 mg/kg of genistein and daidzein, respectively, and 11.3 μg/g total genistein equivalents (TGE, calculated as genistein plus daidzein times 0.1)]. Post-PND22, the same applies for RME1, batch 1028: <10 mg/kg of genistein and daidzein and <10.1 μg/g TGE. All values remained well below the upper limit (325–350 μg/g) suggested by the Organisation for Economic Co-operation and Development (OECD). Female offspring were sacrificed during the diestrus cycle. Animals were weighed and then anesthetized with a cocktail of ketamine (90 mg/kg) and xylazine (10 mg/kg), administered intraperitoneally, in accordance with the Institutional Animal Care and Use Committee guidelines for rat anesthesia. Animals were sacrificed by aortic exsanguination. Tissue and blood samples were collected in a dedicated lab adjacent to the animal facility. A total of 119 rats participated in 5-week ($n = 63$) and 52-week ($n = 56$) experiments, including 50 control rats (25 males and 25 females), 37 BPA0.5 group offspring (19 males and 18 females), and 32 BPA50 group offspring (17 males and 15 females). No adverse events occurred, no animals were excluded, and there were no pre-set criteria for exclusion. All personnel involved in the study were blinded, except for the project director and the staff member recording weights and other data during tissue collection.

### Exposure
Dams were exposed to BPA directly through their drinking water from arrival at GD 3.5 until sacrifice at weaning on PND 22. Bisphenol A (80-05-7, (CH3)2 C(C6H4OH)2, ≥99% purity) (Sigma Aldrich) was dissolved in ethanol and subsequently diluted with well-flushed tap water to defined final concentrations (1% ethanol in the final solutions). We aimed for 0.5 μg BPA/kg BW/day (BPA0.5) and 50 μg BPA/kg BW/day (BPA50). BPA solutions were freshly prepared and replenished, and consumption was measured twice a week. Control dams were given well-flushed tap water containing 1 vol% ethanol (vehicle). The pups were exposed mainly through the placenta in utero and via milk during lactation. However, they may also have been exposed directly through drinking the water during the last days of lactation. The concentration of BPA in the solution was verified at the Division of Occupational and Environmental Medicine in Lund, Sweden. The limit of detection (LOD) for the analysis of BPA was 0.2 ng/mL. The division in Lund serves as a reference laboratory for the European biomonitoring project, specifically the Consortium to Perform Human Biomonitoring on a European Scale (COPHES; www.eu-hbm.info/democophes).

## Microarray of rat bone marrow and probe search

Bone marrow was isolated from the left humerus. Both epiphyses, including the growth plates, were removed, and the marrow was flushed out with a syringe containing ice-cold PBS, followed by collection through centrifugation. The pelleted marrow cells were dissolved in TRI Reagent® (Sigma-Aldrich) and stored at −70 °C. Total RNA was extracted from 24 animals, comprising three groups of 5 females per group and three groups of 3 males per group. Hybridization and scanning were performed simultaneously on the 24-well Affymetrix GeneChip™ Rat Gene 2.1 ST Arrays at the Array and Analysis Facility at Uppsala University. Processing of materials and pre-processing of the raw data (Affymetrix CEL files), according to the standard analysis pipeline at the Array and Analysis Facility. RNA quality was evaluated using the Agilent 2100 Bioanalyzer system (Agilent Technologies Inc., Palo Alto, CA). Two hundred fifty nanograms of total RNA from each sample were used to generate amplified and biotinylated sense-strand cDNA from the entire expressed genome according to the GeneChip™ WT PLUS Reagent Kit User Manual (P/N 703174, ThermoFisher Scientific Inc., Life Technologies, Carlsbad, CA 92008). GeneChip™ ST Arrays (GeneChip™ Rat Gene 2.1 ST Array Plate) were hybridized, washed, stained, and finally scanned with the GeneTitan™ Multi-Channel (MC) Instrument, according to the GeneTitan™ Instrument User Guide for Expression Array Plates (PN 702933, ThermoFisher, Scientific Inc., Life Technologies, Carlsbad, CA 92008 USA). A total of 36,685 array probes with results were annotated with updated NetAffx information at the time of the experiments (in 2018). Statistically significant probes were manually annotated using Ensembl (ensembl.org) in 2024. A search for differentially expressed gene (DEG) functions was conducted using PubMed (ncbi.nlm.nih.gov/gene/) and the Human Protein Atlas (proteinatlas.org).

## Determination of selected markers in rat plasma

Blood from exsanguination was collected in ethylenediaminetetraacetic acid (EDTA)/protease inhibitor-treated tubes and centrifuged (2500×g; 10 min, 4 °C) to prepare plasma. Aliquots were stored at −70 °C. Commercially available enzyme-based kits were used for measuring plasma levels, as follows: alanine aminotransferase/transaminase (ALT, ab105134), aspartate aminotransferase/transaminase (AST, ab13887), and potassium (ab252904), all from Abcam Inc., Cambridge, UK. Meso Scale Discovery multiplex metabolic and inflammation immunoassay panels were used to quantify BDNF, C-Peptide, Ghrelin, GLP1, PYY, Glucagon, Insulin, Leptin, IFNγ, IL1β, IL4, IL5, IL6, IL10, IL13, CXCL1, TNF, CCL2, NGAL, TIMP1, and TSP1 according to the manufacturer's instructions (Meso Scale Discovery, https://www.mesoscale.com).

## Rat lipidomics

Lipids were recovered from 20 µL of plasma using the Folch extraction method[30], with a chloroform:methanol ratio of 2:1. Lipid analysis was completed using the 1290 Infinity UHPLC (Agilent) and QTOF 6540 instrument (YMER) (Agilent)[31] at the Swedish Metabolomics Centre, University of Umeå, Sweden.

## Rat NMR metabolomics

Plasma samples were thawed at room temperature for 15 min, then 90 µL of each sample was mixed with an equal volume of buffer (75 mM sodium phosphate, pH 7.4, 0.08% 3-(trimethylsilyl)propionic-2,2,3,3-d4 acid, 0.1% sodium azide in 20% $v/v$ D2O) in a deep-well plate (2 mL polypropylene, Porvair cat nr 219030). Then shaken at 500 rpm, 12 °C for 2 min in a thermomixer (Eppendorf Comfort). One hundred eighty microliters of each sample was transferred to 4" 3 mm NMR tubes using a SamplePro Tube L liquid handler (Bruker BioSpin). The resulting SampleJet rack was placed in the SampleJet automatic sample changer of a Bruker 600 MHz Avance III HD spectrometer, equipped with a 5 mm room-temperature BBI probe. The samples were kept at 6 °C during preparation and in the SampleJet during data acquisition when not in the spectrometer. Data was acquired and processed using the In Vitro Diagnostics for Research (IVDr) standard operating procedures[32], and metabolites were quantified through automated deconvolution (Bruker Biospin, B.I.QuantPS v2.0.0).

## Human study, EpiHealth

In the EpiHealth cohort, 25,369 random individuals aged 45–75 years were included between 2011 and 2018. Inclusion was performed in two Swedish Cities, Uppsala and Malmö. The 100,685 individuals were invited, resulting in a participation rate of 25%[33,34]. Plasma metabolomics were analyzed in the first 2342 subjects attending the Uppsala site. The relationships between MetS and metabolomics were examined in a cross-sectional manner in this subsample. The mean age in this subsample was 61.1 years (SD 8.4). 50% were females. The mean BMI was 26.5 (3.8) kg/m², and 25% of participants showed MetS. MetS was defined as an individual showing three out of five NCEP/consensus criteria: high blood pressure, increased waist circumference, high fasting glucose, low HDL-cholesterol, and increased triglycerides[35].

Metabolomics (Metabolon Inc., USA) was performed on plasma samples stored at −80 °C. The 500 µL of methanol was added to 100 µL of human plasma for analysis. Samples were prepared using the automated MicroLab STAR® system from Hamilton Company. Lipoproteins and their content were quantified with high-throughput NMR metabolomics (Nightingale Health Ltd, Helsinki, Finland). The 14 lipoprotein subclass sizes were defined as follows: extremely large VLDL, with particle diameters starting from 75 nm and possibly including contributions from chylomicrons; five VLDL subclasses; IDL; three LDL subclasses; and four HDL subclasses. The components quantified within the lipoprotein subclasses included phospholipids (PL), triacylglycerides (TG), cholesterol (C), free cholesterol (FC), and cholesteryl esters (CE). Very few measurements of extremely large VLDL exceeded the detection limit, so this subclass was not used in the current study for further analysis.

## Statistics and reproducibility

Experiments were performed using well-established methods. The raw microarray probe signals were normalized in Affymetrix Expression Console using the RMA-Sketch algorithm to values between 0 and 20, and pre-processed using the IterPLIER method. The background is subtracted from the normalized probe intensities, which generates the relative expression level used here as a surrogate for absolute expression level. The average relative expression level had to be at least 1.5 in one group of the statistically significant comparisons, resulting in the removal of less than 1% of the results. IterPLIER is a pm-gcbg method (background adjusting) that performs a PLIER estimation and iteratively discards non-correlating probe sets. Differentially expressed Gene lists are created using R/Bioconductor (http://www.r-project.org) and the Limma package release 3.6 (no confounding factors included)[36]. No outlier effects were detected by Quality Control (QC) plots using the RMA-sketch version of the robust multi-array average (RMA) method[37]. After processing the arrays, the QC analyses showed no tendencies for outlier samples, and all samples were included in the analysis. Downstream Gene Set Enrichment Analysis (GSEA) was performed using Enrichr (http://amp.pharm.mssm.edu/Enrichr)[38–40] and the Ingenuity Pathway Analysis (Qiagen.com). The animal study was $n = 8–12$/group, a sufficient number for bone tissue and blood analysis in our experience. For the main analysis, the microarray, $n = 5$/group for females and $n = 3$/group for males, more samples in female groups, as their variation is larger than males, from experience.

For human data (EpiHealth), all metabolites were inversely ranked and normalized to obtain normal distributions and a comparable scale. The metabolites (outcomes) were investigated individually against MetS (exposure) using linear regression, with age as a confounder, in a sex-stratified manner. STATA 16.1 was used for analysis, as described in detail previously[33]. When applying an adjustment according to a false discovery rate (FDR) of less than 0.05, the cut-off for significance for the nominal $p$-values provided in the tables should be 0.012 for the MS-based metabolomics and 0.0086 for the NMR-based metabolomics. Thus, after this

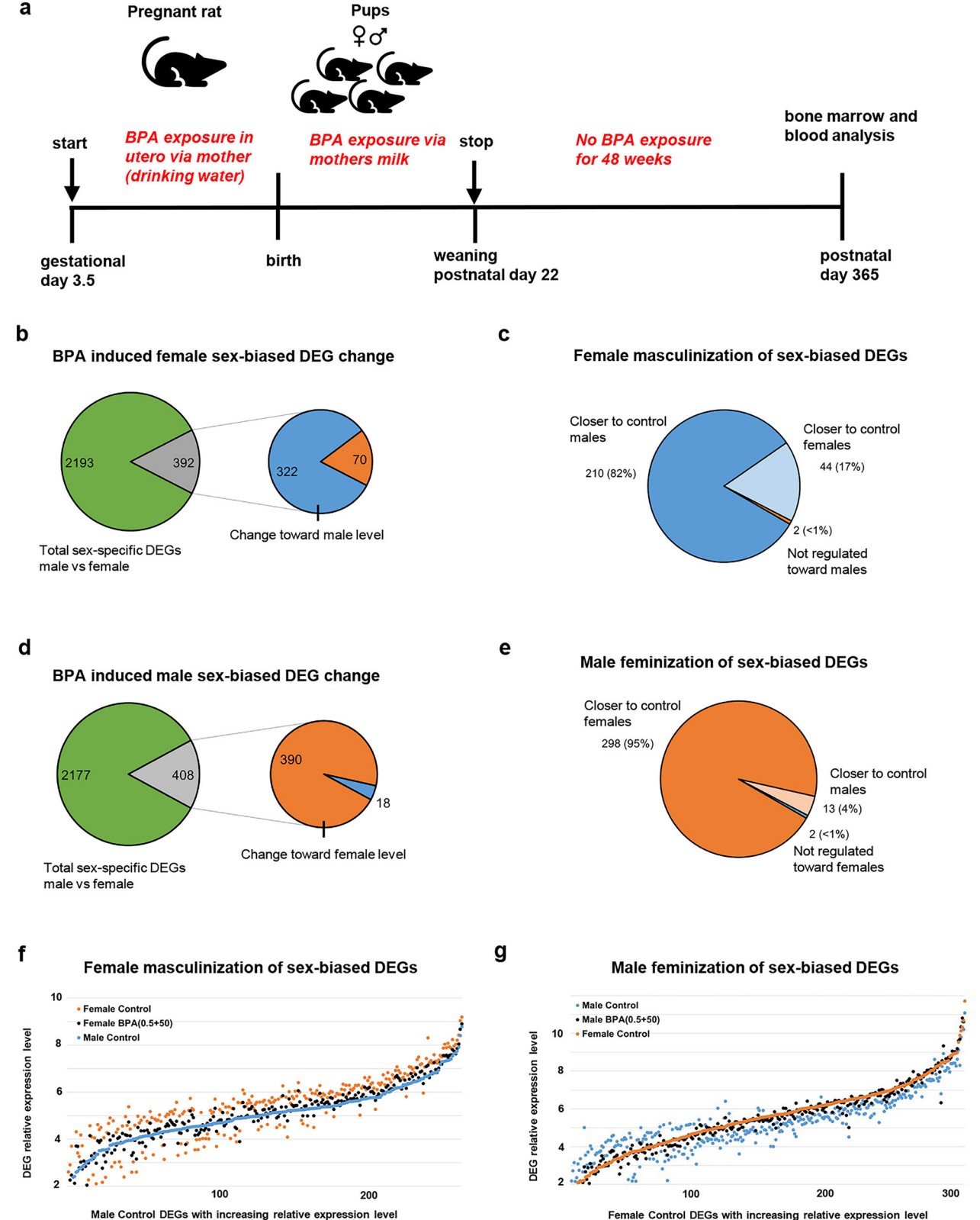

adjustment for multiple testing, most relationships were reported as significant.

Two-way ANOVA was used to analyze rat plasma NMR and lipidomics data to identify sex and treatment differences. Student's *t*-test was used to analyze all other data (except the human analysis described above); in every case, $p < 0.05$ was considered statistically significant without

adjusting for multiple testing to capture all changes in potentially important biological parameters influenced by low-dose developmental BPA exposure.

### Reporting summary

Further information on research design is available in the Nature Portfolio Reporting Summary linked to this article.

**Fig. 1 | Experimental design of F344 rat offspring developmental bisphenol A (BPA) exposure and feminization and masculinization of the bone marrow transcriptome in combined female BPA(0.5 + 50) and male BPA(0.5 + 50) exposure groups. a** Graphical view of the experimental setup, showing the start and stop times of BPA exposure. **b** The total number of sex-biased BPA-induced differentially expressed genes (DEGs) in females (gray). Regulated towards male controls (blue) and female controls (red). **c** The number of female BPA-affected sex-biased DEGs (with identical probe IDs) shifted toward expression levels of male controls (light blue, still closer to female controls) and those closer to male controls (dark blue). **d** Total number of sex-biased BPA-induced DEGs in males (gray).

Regulated towards female controls (red) and male controls (blue). **e** The number of male BPA-induced sex-biased DEGs (with identical probe IDs) shifted toward expression levels of female controls (light red, still closer to male controls) and those closer to female controls (dark red). **f** Relative expression level plot of pie chart from (**c**), female BPA(0.5 + 50) sex-biased DEGs (black) and female control DEGs (red), along with increasing expression levels of male control DEGs (blue). **g** Relative expression level plot of pie chart from (**e**), male BPA(0.5 + 50) sex-biased DEGs (black) and male control DEGs (blue), along with increasing expression levels of female control DEGs (red). BPA-exposed males ($n = 6$), control males ($n = 3$), BPA-exposed females ($n = 10$), and control females ($n = 5$).

**Table 1 | The number of differentially expressed genes (DEGs) from group comparisons in bone marrow microarray analysis of 52-week-old rats developmentally exposed to 0.5 µg/kg BW/day (BPA0.5) or 50 µg/kg BW/day (BPA50) bisphenol A (BPA)**

| All DEGs | Number of DEGs | Up | % | Down | % |
|---|---|---|---|---|---|
| Cont female vs Cont male | 2585 | 2135 | 83 | 450 | 17 |
| BPA0.5 female vs Cont female | 1752 | 963 | 55 | 789 | 45 |
| BPA50 female vs Cont female | 918 | 342 | 37 | 576 | 63 |
| BPA0.5 female vs BPA50 female | 1234 | 786 | 64 | 448 | 36 |
| BPA0.5 male vs Cont male | 843 | 514 | 61 | 329 | 39 |
| BPA50 male vs Cont male | 966 | 523 | 54 | 443 | 46 |
| BPA0.5 male vs BPA50 male | 849 | 554 | 65 | 295 | 45 |
| BPA0.5 female vs Cont male | 3056 | 2497 | 82 | 559 | 18 |
| BPA50 female vs Cont male | 1006 | 748 | 74 | 258 | 26 |
| BPA0.5 male vs Cont female | 3592 | 740 | 21 | 2852 | 79 |
| BPA50 male vs Cont female | 2624 | 1042 | 40 | 1572 | 60 |
| BPA0.5 male vs BPA0.5 female | 4647 | 840 | 18 | 3807 | 82 |
| BPA50 male vs BPA50 female | 1690 | 813 | 48 | 877 | 52 |
| **Sex-biased DEGs** | | | | | |
| BPA0.5 female vs Cont female | 324 | 93 | 29 | 231 | 71 |
| BPA50 female vs Cont female | 289 | 78 | 27 | 211 | 73 |
| BPA0.5 female vs BPA50 female | 91 | 30 | 33 | 61 | 67 |
| BPA0.5 male vs Cont male | 256 | 187 | 73 | 69 | 27 |
| BPA50 male vs Cont male | 270 | 194 | 72 | 76 | 28 |
| BPA0.5 male vs BPA50 male | 73 | 41 | 56 | 32 | 44 |
| BPA0.5 female vs Cont male | 1293 | 1140 | 88 | 153 | 12 |
| BPA50 female vs Cont male | 516 | 435 | 84 | 81 | 16 |
| BPA0.5 male vs Cont female | 998 | 134 | 13 | 864 | 87 |
| BPA50 male vs Cont female | 618 | 95 | 15 | 523 | 85 |
| BPA0.5 male vs BPA0.5 female | 642 | 60 | 9 | 582 | 91 |
| BPA50 male vs BPA50 female | 176 | 46 | 26 | 130 | 74 |

Control (Cont) female $n = 5$, Cont male $n = 3$, BPA0.5 female $n = 5$, BPA0.5 male $n = 3$, BPA50 female $n = 5$ and BPA50 male $n = 3$.
Up represents higher relative expression, and Down represents lower relative expression. The complete DEG lists from each exposure group can be found in Supplementary Data 1.

## Results
### Low-dose BPA exposure shows sex-specific over dose-specific effects on the bone marrow transcriptome

This study examines adult rats 48 weeks after their last BPA exposure, which was via their mother from gestational day 3.5 to postnatal day 22 (Fig. 1a). Microarray analysis of bone marrow generated a list of differentially expressed genes (DEGs), showing expectedly, that the two top-ranked DEGs were from the Y-chromosome (*Ddx3*, $p = 1.1$E-7 and *Eif2s3y*, $p = 5.4$E-6), and with a much higher relative expression level in males when comparing control females and control males (Supplementary Data 1). However, levels were generally higher in females (Table 1). Heatmaps at the individual relative expression levels reveal distinct patterns between groups and similarities within groups, consistent with BPA-induced changes (Supplementary Fig. 1).

As BPA exposure has been linked to female masculinization and male feminization, Table 1 shows a DEG comparison both within each sex and with the opposite sex. This table reveals that the difference in up- and downregulated DEGs is the highest between control males and control females, indicating that BPA exposure reduces the sex differences. Comparing all DEGs (Table 1), the most apparent pattern noticed is the clear trend of the reduced number of DEGs comparing exposed males (or females) with opposite-sex controls, showing that the higher dose (BPA50) is more similar to opposite-sex controls compared to the lower dose (BPA0.5), with a 27–67% reduction of DEG numbers. Additionally, there is a smaller difference between the female and male BPA50 groups compared to the female and male BPA0.5 groups, with a 64% reduction in the number of DEGs. Together, these results suggest trends of female masculinization and male feminization caused by BPA exposure. It further suggests that

**Table 2 | Two-way ANOVA analysis of individual bisphenol A dose (BPA0.5 and BPA50) on the plasma lipidome**

| BPA0.5 | 2-way ANOVA | | | Direction | | Post-hoc Tukey | |
|---|---|---|---|---|---|---|---|
| Lipid name | BPA | sex*BPA | *p*-value | Male | Female | *p*-value | Change |
| LPC | | 16:0 | 0.0451 | | | 0.149 | |
| | | 18:1 | 0.00624 | | | 0.0931 | |
| | | 18:3 | 0.00369 | Down | Up | 0.0487 | mBPA0.5 vs fBPA0.5 |
| | | 20:4 | 0.00534 | | | 0.0590 | |
| PC | 36:6 | | 0.00551 | Down | | 0.0172 | fCont vs mBPA0.5 |
| | | 36:3 | 0.0399 | Down | Up | 0.0083 | fCont vs mCont |
| SM | | 18:1/16:0 | 0.0396 | | | 0.110 | |
| PG | | 47:8 | 0.0219 | Down | Up | 0.0074 | fCont vs mBPA0.5 |
| BPA50 | 2-way ANOVA | | | Direction | | Post-hoc Tukey | |
| Lipid name | BPA | sex*BPA | *p*-value | Male | Female | *p*-value | Change |
| TG | | 44:2 | 0.0203 | | | 0.221 | |
| | | 47:2 | 0.0219 | Down | Up | 0.0408 | fCont vs mCont |

The table shows the lipid identity and significant changes caused by individual developmental BPA-dose exposure. Post-hoc analysis indicates the Direction (up or down) of change, and Change indicates the statistically significant comparison.
Lipidomics: Lysophosphatidylcholine (LPC), Phosphatidylcholine/lecithin (PC), Sphingomyelin (SM), Triacylglycerides (TG), and Glycerophospholipids/Phosphatidylinositol (PG/PI). f (female), m (male), Cont (controls). Exposure groups: BPA0.5 (0.5 µg BPA/kg BW/day) and BPA50 (50 µg BPA/kg BW/day). Number of animals/group: control females *n* = 12, BPA0.5 exposed females *n* = 8, BPA50 exposed females *n* = 8, control males *n* = 12, BPA0.5 exposed males *n* = 8, BPA50 exposed males *n* = 8. Two-way ANOVA, followed by the post-hoc Tukey test.

BPA50 is more efficient than BPA0.5 in this effect. Notably, focusing on only Sex-biased DEGs (Table 1), it appears that both BPA doses in each sex are distinctly more similar to each other than any other group (at least 68-71% fewer DEGs compared to either control), showing that both doses of BPA exposure have similar effects, particularly regarding female masculinization and male feminization.

Compared to same-sex controls, DEGs overlap more (16–30%) within each sex exposed to BPA and have a higher (84–87%) matching expression regulation (up or down) than the same dose overlap (9–18%) with a lower (35–46%) matching expression DEG regulation (Supplementary Table 1). BPA-exposed female groups appear to show more DEG downregulation, whereas exposed males show more upregulation.

Removing DEGs with a fold change (FC) lower than 1.2 does not alter the higher DEG overlap and matching expression regulation (up or down) within each BPA-exposed sex compared to the overlap within a particular dose (Supplementary Table 2). However, it reduces the number of DEGs by 82–89% in females and by 65–70% in males, and the predominant downregulation of DEGs in exposed females and upregulation of DEGs in exposed males are lost. Notably, the average relative expression levels of DEGs in the FC 1.2 cut-off groups are meaningfully reduced, ranging from 56% to 73% of the values obtained without an FC cutoff. Plotting FC against DEG relative expression level clearly shows that using an FC cutoff introduces a bias towards DEGs with lower relative expression levels (Supplementary Fig. 2). To avoid this bias, no FC cutoff was used in the following analysis.

Gene set enrichment analysis using Enrichr reveals the same top pathways (ribosome) in both BPA-exposed male groups (BPA0.5 and BPA50), along with indications of T cell involvement (Supplementary Data 2). Results for BPA-exposed female groups are less clear, although polycomb repressive complex 2, purine metabolism, and T cells appear common in both BPA doses.

**Two-way ANOVA analysis of plasma lipidomic and NMR metabolic profiling in rats exposed to BPA0.5 and BPA50**
We screened blood for metabolic markers using modern assays to determine whether the substantial, distinct, yet modest changes in gene relative expression levels caused by BPA are linked to disturbed metabolism. NMR metabolomics of rat plasma revealed distinct patterns between females and males resulting from BPA exposure. Although the method used is not optimized for rat lipoprotein analysis, we observed that the differences

measured here, comparing rat control females and males, align with published differences measured by NMR analysis comparing women and men[41] (Supplementary Fig. 3).

To compare the metabolic effects between the BPA0.5 and BPA50 doses, we performed a two-way ANOVA analysis on the plasma lipidomics and NMR metabolomics results of exposed rats. Each dose was analyzed separately for a more straightforward interpretation of the effect of the lower dose. Lipidomic results show that the lower dose (BPA0.5) altered levels of four lipids: lysophosphatidylcholine LPC(18:3), phosphatidylcholine/lecithin PC(36:6), PC(36:3) and glycerophospholipids PG(47:8), in exposed males and three: LPC(18:3), PC(36:3) and PG(47:8) in exposed females (Table 2). The higher dose (BPA50) had one altered lipid: triacylglycerides TG(47:2) in both males and females. Notably, both BPA doses in males reduced the levels of all lipids, whereas the levels of all lipids increased in females.

NMR results in males, show that the BPA0.5 dose altered levels of ABA1 (Apo-B1/Apo-A1), V5CH and V5PL (high-density VLDL cholesterol and phospholipids), L6TG, L6CH and L6PL (high-density LDL triacylglycerides, cholesterol and phospholipids), H3PL (HDL phospholipids), Ile and Leu (isoleucine and leucine), and formate (Table 3). All these were increased except for H3PL and formate. In females, NMR results show that the BPA0.5 dose altered levels of V5CH and V5PL (high-density VLDL cholesterol and phospholipids), L6TG, (high-density LDL triacylglycerides), Leu (leucine), and formate (Table 3). All these were decreased except for formate. The BPA50 dose increased in males, while it decreased in females, levels of L2CH, L2FC, and L2PL (low-density LDL cholesterol, free cholesterol, and phospholipids) (Table 3). In addition, BPA50 increased Phe (phenylalanine) levels only in male rats. Notably, BPA exposure consistently had the opposite effect in males compared to females by decreasing lipid levels and increasing lipoproteins and amino acids.

**BPA exposure has extensive effects on sex-biased genes in the bone marrow**
Focusing on the sex-biased effect of BPA, we next investigated the overlap of BPA-induced DEGs with sex-biased DEGs. Between 24% and 38% of BPA-induced DEGs overlap with sex bias (Supplementary Table 3). Remarkably, in BPA-exposed females, 79–88% of these were regulated oppositely compared to the ratio of control females and control males, indicating a high degree of gene regulation towards masculinization. In contrast, 86–89% of BPA-exposed males were regulated similarly to the female/male control

**Table 3 | Two-way ANOVA analysis of individual bisphenol A dose (BPA0.5 and BPA50) on the plasma metabolome**

| BPA0.5 | 2-way ANOVA | | | Direction | | Post-hoc Tukey | |
|---|---|---|---|---|---|---|---|
| NMR name | BPA | sex*BPA | p-value | Male | Female | p-value | Change |
| Main | | ABA1 | 0.0273 | Up | | 0.0347 | mBPA0.5 vs all |
| ApoB | | L6PN | 0.00629 | | | 0.146 | |
| Main V/L/H | | HDPL | 0.0122 | | | 0.176 | |
| Sub VLDL | | V5CH | 0.0482 | Up | Down | 0.00525 | mBPA0.5 vs fBPA0.5 |
| | | V5PL | 0.0135 | Up | Down | 0.0233 | mBPA0.5 vs fBPA0.5 |
| Sub LDL | | L6TG | 0.0427 | Up | Down | 0.0003 | mBPA0.5 vs fBPA0.5 |
| | | L6CH | 0.00124 | Up | | 0.0307 | mCont vs mBPA0.5 |
| | | L6FC | 0.00248 | | | 0.0516 | |
| | | L6PL | 0.00219 | Up | | 0.0413 | mCont vs mBPA0.5 |
| | | L6AB | 0.00630 | | | 0.147 | |
| Sub HDL | | H3CH | 0.0252 | | | 0.322 | |
| | | H3PL | 0.00056 | Down | | 0.0269 | mCont vs mBPA0.5 |
| | | H3A1 | 0.0240 | | | 0.124 | |
| Small Metab | Ile | | 0.00632 | Up | | 0.0031 | mBPA0.5 vs fCont |
| | Leu | | 0.0449 | Up | Down | 0.0464 | fCont vs mCont |
| | Formate | | 0.0403 | Down | Up | 0.0325 | mCont vs fCont |
| BPA50 | 2-way ANOVA | | | Direction | | Post-hoc Tukey | |
| NMR name | BPA | sex*BPA | p-value | Male | Female | p-value | Change |
| ApoB | | L2PN | 0.0123 | | | 0.173 | |
| Sub LDL | | L2CH | 0.0138 | Up | Down | 0.00016 | fCont vs mCont |
| | | L2FC | 0.00942 | Up | Down | 0.00016 | fCont vs mCont |
| | | L2PL | 0.00248 | Up | Down | 0.00016 | fCont vs mCont |
| | | L2AB | 0.0123 | | | 0.172 | |
| Small Metab | Gly | | 0.0389 | | | 0.197 | |
| | Phe | | 0.0200 | Up | | 0.0484 | mBPA0.5 vs fCont |

The table shows the lipoprotein identity and significant changes caused by individual developmental BPA-dose exposure. Post-hoc analysis indicates the Direction (up or down) of change, and Change indicates the statistically significant comparison

NMR (see legend of Supplementary Fig. 3): ABA1 (Apo-B1/Apo-A1), L6PN (high-density LDL, particle number), HDPL (HDL, phospholipids), V5CH/PL (high-density VLDL, cholesterol/phospholipid), L6TG/CH/FC/PL/AB (high-density LDL, triacylglycerol/cholesterol/free cholesterol/phospholipid/Apo-B1), H3CH/PL/A1 (HDL, cholesterol/phospholipid/Apo-A1), L2PN/CH/FC/PL/AB (low-density LDL, particle number/cholesterol/free cholesterol/phospholipid/Apo-B1), Ile (isoleucine), Leu (leucine), Gly (glycine) and Phe (phenylalanine). f (female), m (male), Cont (controls). Exposure groups: BPA0.5 (0.5 µg BPA/kg BW/day) and BPA50 (50 µg BPA/kg BW/day). Number of animals/group: control females $n = 11$, BPA0.5 exposed females $n = 8$, BPA50 exposed females $n = 7$, control males $n = 12$, BPA0.5 exposed males $n = 8$, BPA50 exposed males $n = 8$. Two-way ANOVA, followed by the post-hoc Tukey test.

ratio, indicating a high degree of gene regulation towards feminization. Again, BPA induced a general reduction (65–71%) of relative expressed levels in exposed females, whereas increases (69%) are observed in males (Supplementary Table 3). DEG heatmaps of BPA0.5 and BPA50 doses, presented side by side, reveal a shared pattern in both sexes (Supplementary Fig. 4a, b). More importantly, heatmaps of combined female and male BPA(0.5 + 50) groups reveal that the lower (BPA0.5) and the higher (BPA50) doses share similarity, as they appeared to mix within the hierarchical clustering (Supplementary Fig. 4c, d).

Principal component analysis (PCA) of the microarray result shows a clear separation between females and males; however, within each BPA-exposed sex, the separation is less clear, especially in females (Supplementary Fig. 5). So far, results indicate consistent (opposite) sex-specific effects from BPA-exposure, together with similarities in gene regulation within BPA-doses in the same sex, particularly regarding sex-biased gene expression. Notably, the phenotypic changes observed in these rats are also consistently sex-specific and more pronounced with the BPA50 dose[14,27,29]. Thus, bone stiffness (the ability of a bone to resist bending in a 3-point bending test) and bone marrow fibrosis are the most robust phenotypes observed in these rats, which develop similarly in females with both doses[14]. The phenotypic similarity observed between BPA0.5 and BPA50 is also in agreement with a recent, robust study from the CLARITY-BPA consortium, which employed quantitative unsupervised analysis to demonstrate that

BPA exposure at levels of 2.5 µg/kg BW/day and 25 µg/kg BW/day has a similar effect on rat mammary glands[42]. Here, we chose to explore further the sex-biased similarity between the BPA0.5 and BPA50 doses, increase statistical power, and mimic human exposure by analyzing mixed exposure groups (more relevant for humans) instead of focusing on the specific dose effects observed. Thus, we combined data (BPA0.5 + 50) within each sex, generating one female and one male BPA-exposed group, and compared them with controls. The BPA-exposed female group now showed 1123 DEGs, compared with female controls, and BPA-exposed males showed 1056 DEGs compared to male controls (Supplementary Data 3). There were 156 (14–15%) common DEGs between female and male BPA-exposed groups, and 90 changed in the same direction between females and males. Again, there was considerable overlap between DEGs altered by BPA and DEGs that distinguished control females from males. The BPA-exposed group of females and males now shows 392 (35%) and 408 (39%) DEG overlap with sex-biased DEGs, respectively (Fig. 1b, d). To most straightforwardly compare DEG regulation (up or down) between males and females, the same probe ID DEGs were used, as different probes covering a gene have a tight but varied relative expression level (see examples in Table 4). Along this line, 254 (99%) of 256 sex-biased DEGs in BPA-exposed females were regulated towards levels of male controls (masculinization) when analyzing DEGs with the same probe ID. This number is 50% (68 DEGs out of 136) when using DEGs with different probe IDs

**Table 4 | Differentially expressed genes (DEGs) oppositely regulated by bisphenol A in combined females BPA(0.5 + 50) and males BPA(0.5 + 50) exposure groups**

| Function | DEG | Females BPA Rel. exp. | Control Rel. exp. | p-value | Males BPA Rel. exp. | Control Rel. exp. | p-value |
|---|---|---|---|---|---|---|---|
| Sex determination/ hormone | **Znrf3** | 6.60 ± 0.3 | 6.92 ± 0.1 | 0.021 | 6.64 ± 0.1 | 6.19 ± 0.2 | 0.0028 |
| | Phb2 | 1.84 ± 1.5 | 3.69 ± 0.9 | 0.026 | 11.9 ± 0.1 | 11.5 ± 0.3 | 0.028 |
| Body mass/T2D | Fto | 6.18 ± 0.2 | 6.70 ± 0.4 | 0.0048 | 5.00 ± 0.4 | 4.04 ± 0.3 | 0.011 |
| Immunity | **Nlrc3** | 6.71 ± 0.3 | 7.07 ± 0.2 | 0.018 | 6.77 ± 0.2 | 6.19 ± 0.4 | 0.023 |
| T cells | **Trbv13-2** | 6.06 ± 0.3 | 6.40 ± 0.3 | 0.048 | 5.75 ± 0.3 | 5.24 ± 0.2 | 0.020 |
| Thyroid hormone | Thrap3 | 3,13 ± 0.5 | 3.76 ± 0.1 | 0.014 | 5.63 ± 0.2 | 4.55 ± 0.1 | 0.00015 |
| Lipid metabolism | Ptdss1 | 8.27 ± 0.1 | 8.48 ± 0.2 | 0.012 | 6.16 ± 0.1 | 5.24 ± 0.6 | 0.0285 |
| | Lpcat3 | 7.39 ± 0.1 | 7.58 ± 0.1 | 0.029 | 6.78 ± 0.3 | 6.08 ± 0.4 | 0.020 |
| Autophagy/ proteasome | Becn1 | 5.79 ± 0.4 | 6.29 ± 0.1 | 0.019 | 5.11 ± 0.2 | 4.20 ± 0.2 | 0.00074 |
| | Rnf111 | 5.84 ± 0.3 | 6.30 ± 0.4 | 0.022 | 7.41 ± 0.2 | 6.90 ± 0.2 | 0.0064 |
| | Rnf144b | 3.79 ± 0.4 | 4.44 ± 0.3 | 0.011 | 8.37 ± 0.1 | 8.24 ± 0.1 | 0.016 |
| Vesicle transport/ endoplasmic reticulum | Copb1 | 8.56 ± 0.2 | 8.93 ± 0.2 | 0.0021 | 5.67 ± 0.1 | 5.03 ± 0.2 | 0.00094 |
| | Osbpl2 | 5.44 ± 0.3 | 5.86 ± 0.2 | 0.016 | 5.18 ± 0.5 | 3.80 ± 0.4 | 0.0048 |
| | Stx16 | 6.19 ± 0.3 | 6.62 ± 0.3 | 0.013 | 4.66 ± 0.6 | 3.51 ± 0.2 | 0.015 |
| Transcription/ replication | Ankrd17 | 5.69 ± 0.2 | 6.01 ± 0.2 | 0.013 | 4.42 ± 0.4 | 3.20 ± 0.6 | 0.0057 |
| | Eif4h | 5.30 ± 0.4 | 5.77 ± 0.2 | 0.036 | 5.36 ± 0.3 | 3.85 ± 0.8 | 0.0041 |
| | Tfcp2 | 4.88 ± 0.3 | 5.48 ± 0.4 | 0.0071 | 3.56 ± 0.7 | 2.21 ± 0.5 | 0.021 |

Same probe ID DEG (bold). Average relative expression level (Rel. exp.) is shown as DEG value ± SD. Control female $n = 5$, control male $n = 3$, BPA female $n = 10$, and BPA male $n = 6$. Student's $t$-test.

(Supplementary Data 3). Remarkably, 82% of these 254 female DEGs regulated toward male expression levels had relative expression levels closer to control males than control females (Fig. 1c). BPA-exposed males showed a more pronounced feminization effect as 311 (99%) of the 313 sex-biased DEGs are changed towards female controls, and 298 (95%) have relative expression levels closer to female controls than male controls (Fig. 1e). In addition, of these DEGs overlapping with sex-bias, BPA downregulated 75% in females whereas BPA upregulated 73% in males. Notably, this was especially true in DEGs with a higher relative expression level (Fig. 1f, g). For example, upregulation in males but downregulation in females (Table 4) are DEGs associated with sex determination/sex hormone (Znrf3/Phb2), body mass/T2D (Fto), immunity (Nlrc3), T cells (Trbv13-2), thyroid hormone (Thrap3), lipid metabolism (Ptdss1/Lpcat3), autophagy/proteasome (Becn1/Rnf111/Rnf144b), vesicle transport/endoplasmatic reticulum (Copb1/Osbpl2/Stx16) and transcription/replication (Ankrd17/Eif4h/Tfcp2). According to the extensive feminization and masculinization of the bone marrow transcriptome induced by BPA, the top-ranked DEG in females is upregulation of androgen receptor (Ar, $p = 8.9E-5$), and a gene linked to sex reversal (Hspbap1, $p = 2.0E-5$) in males. In particular, many DEGs for T cell formation and activation, including CD4 helper 1/17 T cells and CD8/exhausted T cells, are upregulated in males exposed to BPA (Table 5). In contrast, females exposed to BPA exhibit downregulation of DEGs associated with overall T cell activity and CD4/CD8 T cells. Along this line, Gene Set Enrichment Analysis using Enrichr on the combined female and male BPA(0.5 + 50) groups indicates T cell involvement as the top-ranked (Supplementary Data 2). Enrichr further suggests a "Ribosome" phenotype in the BPA-exposed male combined group. KEGG analysis of the combined female group suggests: "Maturity onset diabetes of the young" as the top phenotype, although not statistically significant.

## Bone marrow transcriptome bioinformatics predicts clinical chemistry tests for circulating metabolites
The striking results of the extensive feminization and masculinization effects of BPA on the bone marrow transcriptome, which were strengthened by

analyzing combined groups, validate the correctness of these findings. Bioinformatics (Ingenuity Pathway Analysis, IPA) focusing on disease and clinical chemistry indicates no phenotypic overlap between the proposed effects of BPA on females and males (Supplementary Table 4). In BPA-exposed females, IPA suggested cancer as a top disease, and potassium and CRP (C-reactive protein) for clinical chemistry analysis. Precancerous bone marrow lesions have been observed in these BPA0.5 and BPA50-exposed female rats[14], and potassium analysis reveals an elevation ($p = 0.039$) in their blood (Fig. 2a). The microarray results indicate that BPA does not alter the bone marrow expression of the inflammation marker CRP. Screening for blood inflammation markers using multiplex ELISAs (unfortunately, CRP was unavailable for rats) identified a positive correlation to TIMP1 ($p = 0.019$) and TNF ($p = 0.015$) in BPA(0.5 + 50) exposed females only (Fig. 2b, c), similar to what we reported previously for the individual groups[14]. No changes were noted in CCL2, IL10, or CXCL1 in either sex, and signals for NGAL, TSP1, IFNγ, IL13, IL1β, IL4, IL5, and IL6 were below the quantification limits (not shown). For males, IPA suggested an inflammatory response (in line with T cell activation) as the top disease, and altered levels of AST (aspartate transaminase) and ALT (alanine transaminase) were noted for clinical chemistry analysis. Analysis revealed reduced levels of AST ($p = 0.016$) and increased levels of ALT ($p = 0.029$) in the blood of male rats (Fig. 2d, e).

## Plasma lipidomic, NMR metabolic profiling, and ELISA metabolic panel analysis of combined BPA(0.5 + 50) groups
Similar to individual groups (Table 2), significant changes in the blood lipid profile of the rats were observed in the male BPA(0.5 + 50) group, who exhibited reductions in LPC, PC, PG, and phosphatidylinositol (PI) molecules, while no significant effects on TG levels were found (Fig. 3a, c, e, g). Although some tendencies towards increased lipid levels (LPCs and short TGs) were observed in the female BPA (0.5 + 50) group, they only showed a trend towards reduced levels of long TGs (Fig. 3b, d, f, g). Notably, these effects are highly age- and sex-specific, as their 5-week-old siblings, similarly exposed to BPA, show comparable increased levels of TGs in both sexes, and males exhibit no reduction in LPC/PC levels (Supplementary Fig. 6).

**Table 5 | Manual compilation of sex-specific bisphenol A-induced downregulation in females and upregulation in males of T cell-associated differentially expressed genes (DEGs) in combined female BPA(0.5 + 50) and male BPA(0.5 + 50) exposure groups**

| Females, downregulated | | | | T cell Function | Males, upregulated | | | |
|---|---|---|---|---|---|---|---|---|
| DEG | BPA Rel. exp. | Control Rel. exp. | *p*-value | | DEG | BPA Rel. exp. | Control Rel. exp. | *p*-value |
| Cd28 | 5.60 ± 0.4 | 6.10 ± 0.2 | 0.0171 | **Formation** | Cd2 | 6.49 ±0.2 | 5.86 ± 0.4 | 0.0170 |
| Igf1r | 3.56 ± 0.2 | 4.09 ± 0.3 | 0.0036 | **Activation** | Cmip | 6.00 ± 0.3 | 5.46 ± 0.1 | 0.0115 |
| Mr1 | 6.35 ± 0.2 | 6.73 ± 0.1 | 0.0040 | | Foxn1 | 4.65 ± 0.2 | 3.92 ± 0.6 | 0.0314 |
| Nlrp10 | 5.89 ± 0.3 | 6.34 ± 0.3 | 0.0185 | | Gpr65 | 9.60 ± 0.1 | 9.31 ± 0.2 | 0.0217 |
| Tnfrsf11a | 3.92 ± 0.4 | 4.49 ± 0.3 | 0.0201 | | Igf2r | 6.74 ± 0.3 | 6.09 ± 0.1 | 0.0078 |
| | | | | | Ikzf1 | 5.31 ± 0.4 | 4.65 ± 0.1 | 0.0266 |
| | | | | | Itk | 6.37 ± 0.1 | 6.04 ± 0.1 | 0.0113 |
| | | | | | Jaml | 4.07 ± 0.6 | 2.18 ± 1.2 | 0.0134 |
| | | | | | Lcp2 | 11.3 ± 0.1 | 11.0 ± 0.2 | 0.0425 |
| | | | | | Myo1g | 11.0 ± 0.3 | 10.2 ± 0.7 | 0.0474 |
| | | | | | Nfatc1 | 8.51 ± 0.2 | 8.17 ± 0.1 | 0.0488 |
| | | | | | Pik3cg | 9.48 ± 0.2 | 9.08 ± 0.2 | 0.0445 |
| | | | | | RT1-T18 | 4.88 ± 0.2 | 4.27 ± 0.4 | 0.0141 |
| | | | | | Sh2d1a | 4.98 ± 0.3 | 4.36 ± 0.4 | 0.0218 |
| | | | | | Stam2 | 10.0 ± 0.2 | 9.62 ± 0.1 | 0.0424 |
| | | | | | Tagap | 8.94 ± 0.2 | 8.42 ± 0.4 | 0.0361 |
| | | | | | Thrap3 | 5.63 ± 0.2 | 4.55 ± 0.1 | 0.0001 |
| | | | | | Tnfrsf9 | 4.92 ± 0.1 | 4.50 ± 0.2 | 0.0054 |
| Crtam | 5.67 ± 0.4 | 6.11 ± 0.2 | 0.0280 | **CD4 helper** | Cd3g | 6.41 ± 0.1 | 6.06 ± 0.4 | 0.0442 |
| Il7 | 3.36 ± 0.5 | 3.97 ± 0.3 | 0.0377 | **1/17 cells** | Clec4e | 8.12 ± 0.3 | 7.49 ± 0.4 | 0.0285 |
| Lag3 | 4.61 ± 0.3 | 5.08 ± 0.2 | 0.0052 | | Ifnlr1 | 3.63 ± 0.4 | 2.53 ± 0.7 | 0.0172 |
| Mapk4 | 1.36 ± 0.9 | 2.93 ± 1.6 | 0.0284 | | Il16 | 8.31 ± 0.2 | 7.80 ± 0.4 | 0.0262 |
| Rora | 5.82 ± 0.3 | 6.26 ± 0.2 | 0.0142 | | Malt1 | 8.38 ± 0.2 | 7.90 ± 0.3 | 0.0176 |
| Sppl3 | 5.20 ± 0.3 | 5.48 ± 0.1 | 0.0389 | | Map2k6 | 5.57 ± 0.1 | 5.33 ± 0.0 | 0.0130 |
| | | | | | Map3k4 | 7.38 ± 0.3 | 6.87 ± 0.3 | 0.0451 |
| | | | | | Stat4 | 6.96 ± 0.1 | 6.61 ± 0.1 | 0.0005 |
| | | | | | Tnf | 5.84 ± 0.3 | 5.26 ± 0.2 | 0.0165 |
| Tox2 | 5.05 ± 0.3 | 5.51 ± 0.4 | 0.0164 | **CD8/** | *Bhlhe40* | 8.05 ± 0.3 | 7.53 ± 0.1 | 0.0192 |
| Txk | 7.26 ± 0.3 | 7.59 ±0.3 | 0.0487 | **Exhausted** | *Ccl4* | 4.94 ± 0.1 | 3.61 ± 1.1 | 0.0138 |
| | | | | | *Ceacam1* | 10.3 ± 0.1 | 10.1 ± 0.2 | 0.0140 |
| | | | | | *Clec9a* | 5.65 ± 0.2 | 5.19 ± 0.3 | 0.0245 |
| | | | | | *Entpd1* | 9.41 ± 0.1 | 8.89 ± 0.4 | 0.0144 |
| | | | | | *Eomes* | 6.23 ± 0.2 | 5.54 ± 0.1 | 0.0021 |
| | | | | | *Gpr18* | 8.27 ± 0.1 | 7.83 ± 0.3 | 0.0202 |
| | | | | | *Klra5* | 6.30 ± 0.6 | 4.54 ± 0.5 | 0.0026 |
| | | | | | *Klrg1* | 5.92 ± 0.1 | 5.64 ± 0.2 | 0.0487 |
| | | | | | *Pdcd1* | 4.85 ± 0.4 | 4.20 ± 0.3 | 0.0450 |
| | | | | | *Pdcd1lg2* | 5.32 ± 0.4 | 4.37 ± 0.3 | 0.0118 |
| | | | | | *Satb1* | 9.05 ± 0.1 | 8.73 ± 0.1 | 0.0088 |

Average relative expression level (Rel. exp.) is shown as DEG value ± SD. Control female *n* = 5, control male *n* = 3, BPA female *n* = 10, and BPA male *n* = 6. Student's *t*-test.

In the male BPA(0.5 + 50) group, NMR lipoprotein parameter analysis revealed reduced signals for L3PN, L3FC, and L3AB (LDL particle numbers, free cholesterol, and Apo-B1) (Supplementary Fig. 7), which was not observed in the individual groups (Table 3). However, in line with the male BPA0.5 result, male BPA(0.5 + 50) showed reduced levels of H3PL and H3A1 (HDL phospholipids and Apo-A1). The female BPA (0.5 + 50) group showed no significant changes in the lipoprotein pattern compared to the controls (Supplementary Fig. 7), where individual groups indicated reduced levels of VLDL and LDL (Table 3). Instead, the combined BPA(0.5 + 50) females show increases in lactate (*p* = 0.0028), alanine (Ala, *p* = 0.026), phenylalanine (Phe, *p* = 0.037), and tyrosine (Tyr, *p* = 0.038), while reductions in acetate (*p* = 0.0028) and glucose (*p* = 0.012), from NMR metabolite and amino acid analysis (Fig. 4a, b, c, d, e, f) not observed in individual groups. BPA(0.5 + 50) males displayed increases in isoleucine (Ile, *p* = 0.036) but reductions in formate (*p* = 0.049), similar to the male BPA0.5 group (Fig. 4g, h). To

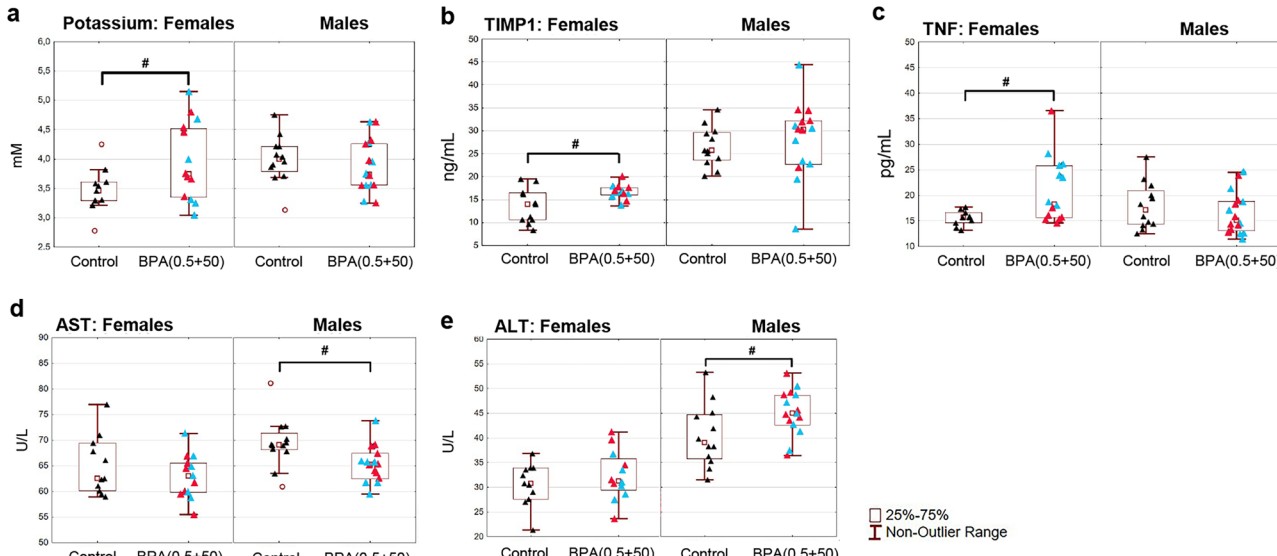

**Fig. 2 | Clinical chemistry tests of plasma in combined female bisphenol A, BPA(0.5 + 50), and male BPA(0.5 + 50) exposure groups. a** Potassium. **b** TIMP metallopeptidase inhibitor 1 (TIMP1). **c** Tumor necrosis factor (TNF). **d** Aspartate transaminase (AST). **e** Alanine transaminase (ALT). Results are expressed as box plots. Blue triangles represent BPA0.5 exposure, red triangles represent BPA50 exposure, and black triangles represent controls. BPA-exposed males ($n = 15$), control males ($n = 12$), BPA-exposed females ($n = 13$), and control females ($n = 11$). #$p < 0.05$ vs control, Student's $t$-test, text for exact $p$-values.

support the NMR results, protein analysis (multiplex ELISA) of a panel of metabolic regulators in plasma revealed increased levels of insulin ($p = 0.028$) in BPA(0.5 + 50) females only (Fig. 4i). No effects on Leptin, Grehlin, Glucagon, or GLP1 were seen, and BPA(0.5 + 50) males showed no changes.

### Plasma metabolic profiling in women and men with MetS reveals similarities and differences

To determine if the BPA-induced metabolic disturbances observed in rats also exist in humans, we conducted a similar omics analysis in a Swedish population-based study involving 2342 individuals (EpiHealth). NMR profiling reveals highly similar metabolic patterns of MetS in women and men (Supplementary Data 4). Top-ranked lipoprotein changes are reductions in mainly cholesterol species, as well as phospholipids, primarily in HDL particles (medium/large/very-large sized) and LDL (medium/large), and very low-density lipoprotein (VLDL) (small/medium), accompanied by a reduction in Apo-A1. This is followed by increases in TG content in all types of lipid particles, together with high levels of leucine, glucose, Ile, Apo-B/Apo-A1 ratio, Apo-B, Ala, valine, Phe, Tyr, and lactate, but low levels of glutamine, citrate, and acetate. As we have observed sex-specific effects in BPA-exposed rats, we listed the top 10 altered parameters in humans with MetS, which are significant in women but not men, and vice versa (Supplementary Table 5). This analysis revealed that women with MetS tended to increase cholesterol species associated with VLDL/LDL, whereas men showed a reduction in cholesterol species associated with VLDL/IDL/LDL. In addition, MetS was associated with decreased TG levels (VLDL-associated) only in women.

Plasma lipidomics from the human study revealed that the top six most significant changes were identical in both women and men with MetS (Supplementary Data 5). These were reduced levels of two glycerophosphocholine species and increases in glucose, glutamate, and two diacylglycerols, followed by reductions in LPC species and increases in urate, phosphatidylethanolamine species, and Ile. Notably, parameters differentiating women and men (significant in one sex only) with MetS were for women with increased testosterone metabolite levels. In contrast, men showed low testosterone metabolism (pregnenediol sulfate) and signs of increased NADH metabolism (valerate, pentanoate, kynurenate) (Supplementary Table 6).

### Low-dose developmental BPA exposure in rats results in a plasma metabolic profile that overlaps with human MetS

Finally, the BPA-induced metabolic profile observed in male rats significantly overlapped with the human MetS, where six of eight parameters aligned well (Table 6). One parameter (concentration of large-sized LDL particles) showed the exact relative change, i.e., higher in women and female BPA-exposed rats compared to men and male BPA-exposed rats. However, compared to controls, the values of women were increased, whereas those of male rats were reduced. The levels of PI species appeared opposite in BPA-exposed male rats (decreased) compared to humans with MetS (increased). Five of the six BPA-induced changes detected by blood omics in female rats aligned well with human MetS (Table 6). The parameter not aligning with human MetS was glucose, typically elevated in humans with MetS but reduced in BPA-exposed female rats. Together, 7 of 8 (87%) of the total BPA-induced blood-omics changes identified using an unbiased approach in male rats overlap with a top-ranked human MetS signature, and the corresponding number for female rats is 5 of 6 (83%).

### Discussion

The main finding of the present study is that a BPA dose approximately eight times lower than a temporary EFSA human tolerable daily intake of 4 µg/kg BW/day showed extensive female masculinization and male feminization effects on the bone marrow transcriptome later in life, similar to a 100-times higher dose, which was considered safe in 2015. The results manifested as at least 210 (8%, in females) and 298 (12%, in males) sex-biased genes changing in BPA-exposed animals to expression levels closer to those of the opposite sex than to same-sex controls. BPA exposure repressed sex-biased genes in females but amplified such genes in males. This affected many essential genes regulating T cells, which are generally more abundant and active in females. However, after developmental BPA exposure, gene expression levels were reduced in females and increased in males. Together, microarray, gene ontology, and blood analysis/omics indicated an immunodeficiency/cancer phenotype with a hypometabolic state in BPA-exposed female rats. In contrast, BPA-exposed male rats exhibited a MetS/autoimmunity phenotype characterized by a hypermetabolic state[43]. Compared to similar blood omics analyses in a population-based cohort, male and female BPA-exposed rats exhibit a substantial metabolic pattern that overlaps with a human MetS phenotype, albeit sex-specific.

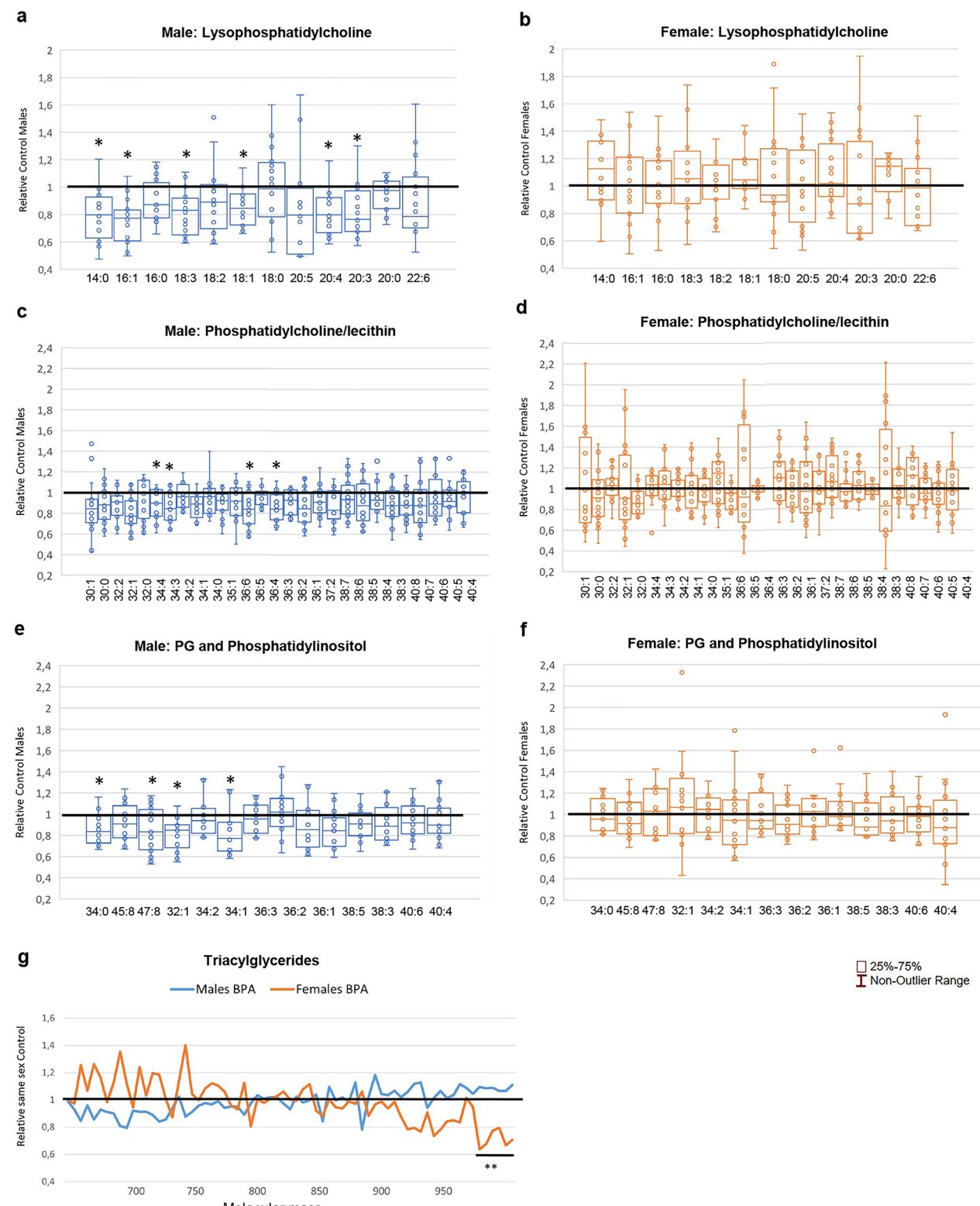

**Fig. 3 | Rat plasma lipidomic profiling in combined female bisphenol A, BPA(0.5 + 50), and male BPA(0.5 + 50) exposure groups.** Males (**a**, **c**, **e**, **g**) and females (**b**, **d**, **f**, **g**). **a**, **b** Lysophosphatidylcholine. **c**, **d** Phosphatidylcholine/lecithin. **e**, **f** Glycerophospholipids (PG) and phosphatidylinositol molecules. **g** Triacylglycerides. Results expressed as box plots (or line, **g**) of exposed females/ males relative to female/male controls. The black line represents value when there is no difference compared to same-sex controls. BPA-exposed males ($n = 16$), control males ($n = 12$), BPA-exposed females ($n = 16$), and control females ($n = 12$).
$^{*}p < 0.05$ and $^{**}p < 0.01$ vs control, Student's $t$-test, Supplementary Data 7 for exact $p$-values.

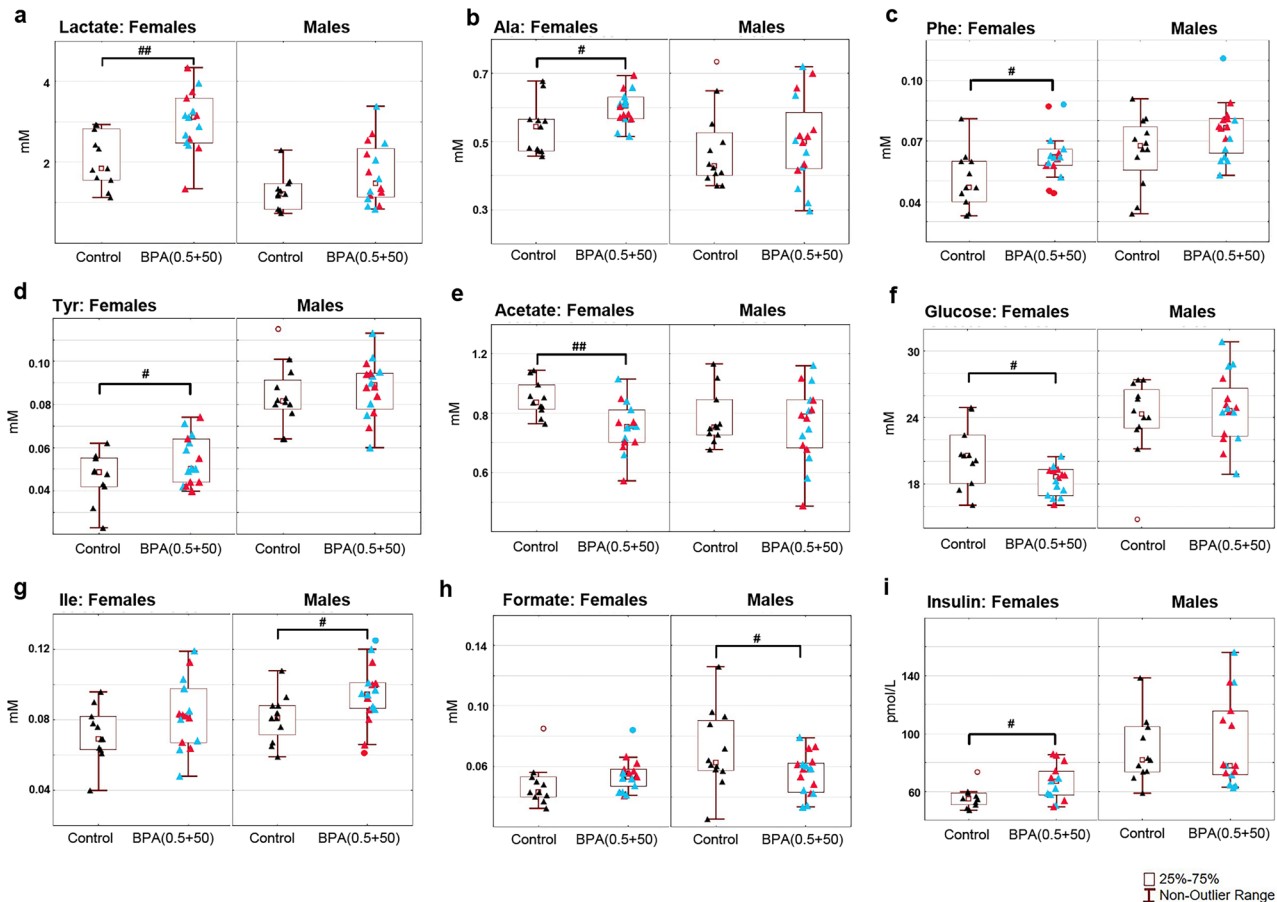

**Fig. 4 | Rat plasma NMR metabolite profiling and ELISA insulin levels in combined female bisphenol A, BPA(0.5 + 50), and male BPA(0.5 + 50) exposure groups.** NMR results: **a** Lactate. **b** Alanine. **c** Phenylalanine. **d** Tyrosine. **e** Acetate. **f** Glucose. **g** Isoleucine. **h** Formate. ELISA insulin: (**i**). Results are expressed as box plots. Blue triangles represent BPA0.5 exposure, red triangles represent BPA50 exposure, and black triangles represent controls. BPA-exposed males ($n = 16$), control males ($n = 12$), BPA-exposed females ($n = 15$), and control females ($n = 11$). #$p < 0.05$ and ##$p < 0.01$ vs control, Student's $t$-test, text for exact $p$-values.

BPA consistently changed DEG and blood metabolite levels in opposite directions in males and females. This aligns with sex-specific effects previously observed in these rats, characterized by phenotypes of reduced bone stiffness, bone marrow fibrosis, increased body weight, and epididymis inflammation. The higher BPA dose (although considered low) exhibits a stronger phenotype[14,27–29]. Thus, as the lower BPA dose shows fewer and weaker phenotypes, the more numerous plasma changes induced by the lower BPA dose, identified by two-way ANOVA analysis of rat blood metabolites, may reflect an earlier disease stage rather than an increased hazard. The observed phenotypic similarity within both doses aligns with a recent study from the CLARITY-BPA consortium, which employed quantitative unsupervised analysis to demonstrate that BPA exposure at 2.5 μg/kg BW/day and 25 μg/kg BW/day has a similar effect on rat mammary glands[42]. In addition, only the lowest BPA dose tested, 25 μg/kg BW/day, similar to our higher dose (BPA50: 50 μg/kg BW/day), has been associated with masculinization of females and feminization of male rats[10,11]. To explore these signs of sex-biased similarity between BPA0.5 and BPA50 within each sex, increase statistical power, and simultaneously adopt a more similar approach to the variable human exposure (more clinically relevant) of investigating the effects of low-dose BPA exposure, we combined data within each sex, generating one female and one male BPA-exposed group, and compared them with controls. Notably, among the DEGs in BPA-exposed offspring that overlap with sex bias, 82% and 95% (in females and males, respectively) exhibited expression levels more similar to those of the opposite sex. In females, 75% were downregulated by BPA, whereas 78% were upregulated in males. Together, these observations convincingly

demonstrate that developmental low-dose BPA exposure has a lasting impact on the bone marrow transcriptome.

In line with this, females have higher hematopoietic stem cell activity, and estrogen is an overall activator of immune cells, while androgen is a suppressor[44]. Bone marrow expression levels were generally higher in control females than in males. Masculinization and a general decrease in gene expression in BPA-exposed females are consistent with the top-ranked DEG being associated with the androgen receptor (*Ar*, $p = 8.9E-5$), for which transcription was upregulated, and the highly upregulated (>5-fold) transcriptional repressor *Zfp644* ($p = 0.00037$). In males, the top-ranked DEG from BPA exposure was increased expression of a ribosomal protein gene (*Rplp2*, $p = 1.3E-5$), which implicated increased cell proliferation and was in line with the overall increased gene expression in male bone marrow. Along this line, bone marrow hypercellularity has been described in 2-year-old male offspring developmentally exposed to BPA in the CLARITY-BPA program[8]. In addition, according to extensive transcriptome feminization in BPA-exposed male rats, the second-ranked DEG maps to *Hspbap1* ($p = 2.0E-5$) and is linked to sex reversal activity[45].

Bioinformatics on the female DEG pattern from low-dose BPA exposure indicated cancer as a top disease, with potassium and inflammation for clinical chemistry tests. It is well established that exposure to another xenoestrogen, diethylstilbestrol, during pregnancy induces cancer in daughters more than sons[46], and screening for carcinogenic properties of BPA in mice showed that females were more likely to develop tumors[47]. It is well known that immunosuppression, as indicated by the reduced expression of critical T cell genes in BPA-exposed females, increases the risk of

**Table 6 | Table of all significant changes in combined female (bisphenol A) BPA(0.5 + 50) and male BPA(0.5 + 50) exposed rats from plasma metabolic profiling (NMR) and lipidomics (Lipi), with corresponding results from a similar analysis of a population-based study of humans with MetS**

| **Male** | | | | |
|---|---|---|---|---|
| Signif. change | Rat NMR BPA | Rat Lipi BPA | Human NMR MetS | Human Lipi MetS |
| L-sized LDL conc. | Reduced | --- | ns | --- |
| L-sized LDL FC | Reduced | --- | Reduced | --- |
| L-sized LDL AB | Reduced | --- | --- | --- |
| M-sized HDL PL | Reduced | --- | Reduced | --- |
| M-sized HDL A1 | Reduced | --- | --- | --- |
| Isoleucine | Increased | --- | Increased | Increased |
| Formate | Reduced | --- | --- | --- |
| LPC | --- | Reduced | --- | Reduced |
| PC | --- | Reduced | --- | Reduced |
| PG | --- | Reduced | --- | Reduced |
| PI | --- | Reduced | --- | Increased |
| **Female** | | | | |
| Signif. change | Rat NMR BPA | Human NMR MetS | | |
| Tyrosine | Increased | Increased | | |
| Phenylalanine | Increased | Increased | | |
| Alanine | Increased | Increased | | |
| Glucose | Reduced | Increased | | |
| Lactate | Increased | Increased | | |
| Acetate | Reduced | Reduced | | |

*ns* insignificant, --- unavailable, *L* large, *M* medium, *LDL* low-density lipoprotein, *HDL* high-density lipoprotein, *FC* free cholesterol, *AB* Apo-B1, *PL* phospholipid, *A1* Apo-A1, *LPC* lysophosphatidylcholine, *PC* phosphatidylcholine/lecithin, *PG* glycerophospholipid, *PI* phosphatidylinositol.
—rat data: BPA-exposed males (*n* = 16), control males (*n* = 12), BPA-exposed females (*n* = 16), and control females (*n* = 12)—human data: a population-based sample of 2342 individuals.

tumor and metastasis development. We have previously demonstrated the presence of precancerous bone marrow lesions and signs of chronic inflammation in the female BPA-exposed rats of the present study[14]. Here, these females show elevated levels of potassium and TIMP1, which in some studies have been associated with increased risk of cancer in humans[48,49]. In BPA-exposed males, the DEG bioinformatics pattern suggested an inflammatory response as the top disease, accompanied by altered levels of AST and ALT, indicating changes in clinical chemistry. Along this line, BPA-exposed males showed upregulation of DEGs controlling immunity cells (mainly T cells) with reduced AST levels and increased ALT levels. This finding aligns with a CLARITY-BPA study, which shows that T cell activation increased only in aged males after low-dose exposure[50] and that developmental BPA exposure upregulates ALT, but not AST, in rat offspring[51]. Notably, ALT levels were recently, independently, and positively associated with hypermetabolism in subjects with non-alcoholic fatty liver disease and type 2 diabetes[52].

Signs of altered bone marrow adipose tissue are present in BPA-exposed offspring, as females exhibit reduced levels of a visceral white adipose tissue (WAT) marker (*Gpc4*, *p* = 0.014). In contrast, males exhibit an increase in a subcutaneous WAT marker (*Tbx15*, *p* = 0.044). We have previously shown reduced *Scd1* expression in the adipose tissue of 5-week-old male rats exposed to BPA[26]. Here, we find that their 52-week-old BPA-exposed male siblings still exhibit reduced *Scd1* expression (*p* = 0.043). *Scd1* deficiency increases energy expenditure in mice[53,54]. Along these lines, in

humans, bone marrow activation is more frequent in men and is associated with an earlier onset of MetS and atherosclerosis[55]. We have previously observed an inverse correlation between energy expenditure and blood lipids in humans[56]. Here, mainly male BPA-exposed rats exhibited a disturbed lipoprotein profile, characterized by a specific reduction in signals for LDL cholesterol and phospholipid HDL. Cholesterol is a precursor to sex hormone synthesis, and, similar to male rats exposed to BPA, LDL cholesterol was negatively associated with MetS in men. In contrast, women exhibited a positive association between LDL cholesterol and MetS. In harmony with this, lipid transport appears to be mainly altered in males exposed to BPA. They also developed reductions in LPC, PC, PG, and PI species but minimal changes in TG levels. Reduced blood lipids and increased metabolic rate are phenotypes of increased thyroid hormone activity. Along this line, a thyroid hormone receptor-associated protein (*Thrap3*, *p* = 0.00015), linked to T cell activity and type 2 diabetes, is one of the top upregulated genes in males. Notably, in humans with MetS, women showed increased androgen metabolism. In contrast, men were linked to reduced levels of androgen metabolism, further establishing female masculinization and male feminization as essential contributors to the development of MetS/obesity in humans[57]. In line with this, the top-ranked metabolites for MetS in humans are the same for men and women.

In contrast to males, female BPA-exposed rats exhibited altered levels of small molecule metabolites (lactate, Tyr, Phe, Ala, acetate), similar to the changes observed in humans with MetS. The exception is glucose levels, which were lower in BPA-exposed females and unaltered in male rats. The top-ranked downregulated DEG in females is a *Ces2* member (*p* = 0.00029), a key protector of metabolic disease involving glucose metabolism[58–60]. Low glucose levels may be associated with increased activity of Ar/testosterone or may indicate the presence of tumors or cancer[61]. A further sign of cancer in the BPA-exposed females is their selectively reduced levels of long TGs, recently shown as indispensable for tumor growth[62]. BPA-exposed females in the present study also exhibited increased insulin levels, a hallmark of MetS. In humans, sex hormones are important regulators of glucose metabolism, and hypoglycemic hyperinsulinemia is more common in women[63]. The female BPA-exposed rats have shown signs of elevated blood testosterone[14], and together with elevated TNF and suppressed T cell function observed in the present study, mimic masculinization in women[64]. In addition, the upregulation of *Ar*, observed in this study, adds to previously published data suggesting that BPA may influence the development of polycystic ovary syndrome (PCOS)[65].

The literature on how low-dose BPA exposure worsens health suggests that BPA induces epigenetic changes during the sensitive fetal period, which can persist across generations[66]. These changes appear to impact the function of many organs, and the present work provides further insight into the mechanism underlying the effect on the bone marrow transcriptome and blood metabolites. BPA exposure appears to disrupt the subtle sex-specific differences in the highly evolved and tightly controlled regulation of gene expression, which occurs over a prolonged period, disrupting the specific female and male metabolisms towards a MetS phenotype. Our results indicate that bone marrow T cell function is altered in opposite directions in males and females exposed to BPA. As bone marrow metabolism has been shown to reflect overall body metabolism and regulate immune homeostasis in peripheral tissues[22], it suggests that disturbed bone marrow function may significantly contribute to the metabolic changes induced by BPA exposure.

The human data were used to find similarities between metabolic changes in human MetS and BPA exposure in rats. A limitation is that we do not have measurements of BPA in the human sample. The human data are also limited to a middle-aged Swedish population.

In conclusion, BPA induced considerable sex-specific gene repression in females but stimulation in males, particularly associated with T cells. Along this line, EFSA has recently lowered the human TDI by 20,000-fold based on BPA's critical effect on T cells[67]. Our results indicate that BPA induces a hypometabolic/cancer state in females and a hypermetabolic/autoimmune state in males. Notably, most blood omics metabolic markers observed in BPA-exposed rats were also found in human MetS. This study

further highlights the bone marrow transcriptome as tightly controlled, reflecting the overall body metabolism, and highly sensitive to BPA. Notably, as the bone marrow appears to play a crucial role in pregnancy, it may be worthwhile to investigate DEGs from this study to gain insight into how BPA affects female fertility[68,69]. Furthermore, it would be of interest to explore a potential connection between BPA exposure and the sharp rise in gender incongruence[70].

## Data availability

The source data are provided for Fig. 2 in Supplementary Data 6, for Fig. 3 in Supplementary Data 7, and for Fig. 4 in Supplementary Data 6 and 8. The rat microarray data discussed in this publication have been deposited in Annotare—EMBL-EBI (https://www.ebi.ac.uk › annotare) and are accessible through number E-MTAB-15555. Swedish law does not permit human health data to be available in the public domain. However, the data could be obtained for research purposes upon a reasonable request from the authors.

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

## Acknowledgements

The authors would like to thank the Swedish Research Council for Environment, Agricultural Sciences, and Spatial Planning (FORMAS) for providing funding (216-2012-475). The authors also wish to acknowledge Britt-Marie Andersson and Ann-Marie Gustafson for excellent technical assistance. Array and Analysis Facility, Department of Medical Sciences at Uppsala Biomedical Center (BMC), Husargatan 3, 751 23 Uppsala. Support from NBIS (National Bioinformatics Infrastructure Sweden), the Swedish Metabolomics Centre, the University of Umeå, and the Swedish NMR Centre at the University of Gothenburg is gratefully acknowledged. The Swedish Research Council funded the population-based EpiHealth study as part of a strategic research area (SFO).

## Author contributions

T.L. designed the study, wrote the manuscript, performed tissue collection, extracted RNA, and performed animal blood analysis except for NMR and Lipidomics. P.M.L., M.H.L., and L.D. designed and performed the animal

exposure study, collected tissues, and contributed to the writing of the manuscript. L.L. performed the human analysis part (EpiHealth) and contributed to the writing of the manuscript. H.M. participated in writing the manuscript.

## Funding

## Competing interests
The authors declare that they have no competing financial interests.
