## [Transparent Peer Review file · Communications Medicine]

Developmental low-dose bisphenol A exposure leads to extensive transcriptome female masculinization and male feminization later in life

Corresponding Author: Dr Thomas Lind

Version 0:

Reviewer comments:

Reviewer #1

(Remarks to the Author)

Thanks to the editor for the invitation.

This study investigated sex-specific effects of BPA on bone marrow transcriptome and blood metabolome. However, the findings lack validation, the data results are not enough to support the conclusion, and the quality of the manuscript needs to be improved. Overall, this study is not enough to be published in a high-level journal.

The experimental design is puzzling. The entire study, including genetic testing and analysis, as well as the examination of metabolic data in the population, was taken from published articles and appears to be a recombination of previously published data.

The language of the manuscript needs to be improved.

Abstract. The abstract section needs to reorganize the logic. And it is recommended that the abstracts do not cite references. Results. Both in the descriptive text of the results and in the graphical presentation, need to be further modified. The description of the results requires a summative description, streamlining, and highlighting the results. The presentation of the figures or tables suggests highlighting the point rather than listing all the results and leaving the reader looking for the resulting data.

Methods. The experimental methods of important experiments need to be detailed, for example, information on sample collection at the end of animal exposure, information on statistical analysis of data.

Reviewer #2

(Remarks to the Author)

The authors use differential expression analysis and GSEA to provide evidence for significant effects from low-dose BPA exposure. Although this study includes important results on BPA exposure, there are several issues that have to be addressed before I can recommend the paper for publication. In particular, some statistical methods have to be described in more detail (see my comments below), and some figure and table captions contain insufficient information to fully understand what is displayed.

Major:

- Limma: the authors do not report if any confounding factors were included.
- Lines 901-902: No outlier effects were revealed by Quality Control (QC) plots -> which software / package / method did you use for the quality control?
- Figure 3, 4, 5, Table S4, Table S5: are the p-values displayed adjusted for multiple hypothesis testing?
- Supplementary Table S3: (adjusted) p-values are not reported.

Minor:

- Not common to add references in the abstract.
- Line 130: change -> changed
- Line 201: patters -> patterns
- Line 402: indicate -> indicates
- Line 775-776: time-mated -> do you mean age-matched?

- R/Bioconductor and limma: please mention which version you used.
- The authors report adjusted p-values. Which procedure was used? Benjamini-Hochberg?
- Captions extended data Tables and Figures, Fig 4 a-d, Fig 5: please explain from which statistical method these results are.

Reviewer #3

(Remarks to the Author)

Thank you for the opportunity to review COMMSMED-23-0432-T by Dr. Lind et al. Expanding previous work which highlighted the sensitivity of bone to BPA exposure, the enclosed study demonstrates that BPA developmental exposures (relevant to human exposure) alters the BM transcriptome and blood metabolome. Interestingly, the results show that BPA afflicts bone profiles/metabolic profiles in a sex-specific manner which may have implications regarding BPA contributions to immune cell dysfunction and human metabolic syndrome. Overall, the study is well written and of high importance for toxicology and risk assessment. The results are compelling; however, I have significant reservations regarding the interpretation of the results due to the preliminary nature of the transcriptomic analysis and missing data that supports sex-dependent phenotypes associated with BPA developmental exposure observed at 52 weeks.

Major concerns.

1. Assessments at 52 weeks likely reflect phenotypic differences as an outcome of developmental exposure vs. perturbations due to BPA exposure. This point is sometimes lost. What are the effects of BPA at each of these concentrations at birth, PND22, etc. prior to these assessments? Were any cellular/ molecular assessments completed during exposure period to understand how effects of BPA manifest into differences observed at 52 weeks? Could the authors provide supplemental experimental data or at least a summary of expected differences at these earlier timepoints? Could the authors add a schematic which illustrates the experimental design, emphasizing examination of BM and metabolic profiles at 48 weeks after last exposure (perinatal) and highlights differences in profiles between the two testing doses.
2. The authors have provided a preliminary analysis of the transcriptomic data. I would encourage using statistical tests and methods to optimally examine sex-dependent effects in the context of BPA testing concentrations. Interpretation of probe intensities is misconstrued. Need to better justify separation of male/female or combining of BPA dose groups.

Specific Comments

Transcriptomic analysis: Overall, the analysis is preliminary and the authors do not provide rationale to support performed analyses or conclusions. I would suggest using different statistical approaches to examine sex-dependent differences in the context of BPA exposure. For example, would PCA plots using all samples suggest stratification of samples by sex (in exposed and controls), justifying analysis to look at sex-specific changes? Provided Table 1 and Figure 1 provide suggest differences across groups, but poor indication of overall trends.

The number of DETs is very large for many of the comparisons. Did you assess with more stringent criteria to see if your conclusions will still be the same. For example, it would be interesting to incorporate more-restrictive p-value and FC. Is the p-value <0.05 corrected for multiple testing?

Data could be cleaned up. Analysis still includes probes IDs not linked to DET (see Table 2), potentially non-informative information. Were lowly expressed transcripts removed from the analysis? Why were analyses completed on the gene level? Or, can the authors provide information about unique genes vs. transcripts.

Generalized linear models could be applied to assess dose-dependent changes (in both/either males and females) to identify DETs, while enabling robust power (using all data). More complex GLMs could be used to identify sex-specific changes that are differentially expressed between sexes in the context of BPA exposure (interactions).

I find it very confusing that later on in the analysis (Figure 2) that the authors combine DETs from 0.5 and 50ug/kg groups. How is this justified based on Table 1/Figure 1 data? There seems to be very limited overlap in these profiles. Did BPA genes (0.5 vs. 50) change in a similar manner with BPA in males or females? Current justification (line 138) is due to large differences in sex vs. exposure (which is expected), and "the high overall similarity in DET regulation". As for the latter point, where is this shown? Figure 1 suggests limited overlap (VENN; 5-10% overlap in males or females).

Overall, could you provide PCA plots and heatmaps that highlight differences/similarities among groups? Table 1 and Figure 2/3 provides number and comparisons, but difficult to see overall relationships.

3. Justification of metabolic analysis. In its current format, the authors do not provide interpretation of the data; more of a data dump. Can the authors categorize results to provide a clear picture in how early BPA developmental exposures alter metabolic/lipidomic signatures later on in life?

4. Increased description of quantitative changes is needed. The authors appropriately emphasize significance using p-values, however, could provide more quantitative data regarding fold change; change in abundances to understand biological relevance of altered endpoints.

Introduction

Line 61. Please define "huge". Can you note diversity and number of studies in meta-analysis?

Line 88. Is there evidence of specific mechanisms demonstrating how altered endocrine/nuclear receptor activity can lead to changes in bone?

Line 269. Heatmaps which show changing/revert expression changes would greatly help support this point.

Line 273. Heatmap would greatly support this.

Line 898. What is a pm-gcpg method?

Line 906. What is the cutoff criteria for DETs?

Tables/Figures

Table 1. What is the cutoff criteria for DETs? Any information regarding fold differences?

Table 2. In using microarray probes, I would hesitate calling these "expression levels". Probes intensities represent relative expression. Would it be possible to log2 adjust to control group (average or male). What is the significance of these DETs?

Would make more sense to separate control data (into supplemental table) and focus on BPA effects. Using GLM and interaction terms would enable the ability to identify commonly altered genes and sex-specific.

Table 3. Why are dose groups combined? Was this a separate analysis using limma to look at dose-dependent effects? What are the range of p-values signifying? Where are all of the other categories?

Table 4. Could this be shown in a heatmap? Sense of FC and significance should be provided or described.

Table 5. What are the importance of these different groups? What are the range in values? What is significant?

Figure 1. Why are titles b-e labeled masculinization or feminization? Panels b-e are supplemental, difficult to read and interpret. Smaller dots would be appreciated. Probe values should be adjusted and heatmaps could be used to reflect differences among groups. I would encourage using a method which enables statistical power to identify BPA effects across doses; unless PCA plots clearly show a divergence among dose groups. Probe values do not necessarily correspond to abundance/expression levels.

Figure 2. Why are does groups combined after showing differences in Fig. 1? Do heatmaps illustrate sex-expression reversion?

Figure 3 and 4. Why are these data combined? What is the reasoning for these diverse measurements? Box plots showing single values would be helpful to see data.

Figure 4a-d. What are the units? Why are these measurements grouped together?

Figure 5. PCA? Difficult to read. Is this supplemental?

Version 1:

Reviewer comments:

Reviewer #2

(Remarks to the Author)

The authors have addressed my comments sufficiently.

Reviewer #3

(Remarks to the Author)

Thank you for providing additional details and clarifying the importance of the results. The manuscript is much improved and represents tremendous work; however, there are still concerns regarding flow, rigor and evidence supporting the conclusions.

Introduction.

While the modifications have added some clarity, the text is difficult to follow. I would suggest rewriting and checking that each paragraph is clearly providing the context that you are trying to convey to the reader. There are flow issues. Would it make sense to start with Paragraph 2 (line 86) discuss importance of bone and lack of knowledge in toxicity studies? Moving to BPA (general, paragraph 2) and then low-dose effects in bone (paragraph 3)? Then, lead into overview of study (paragraph 4)? The current Paragraph 1 is very difficult to read. Key concepts should be broken into separate paragraphs: 1) BPA (general) and low-dose effects; 2) sensitivity of bone and lack of knowledge at low dose.

It should be very clear that previous work and the model builds on previous published studies (phenotype at PND35, refs 13, 26, 27) and this extends this research. Can this be integrated more clearly into the last paragraph?

Methods

Line 166: Please verify. Were steps in the process randomized (RNA isolation, scanning of microarrays)? Were males and females processed at the same time?

Results

Line 275: While the authors have stated that this is preliminary, are there concerns that the interpretation of this microarray data is largely based on probes not assigned to genes of known function. Also, includes untranslated RNA (small, pseudo, miR) and many non-annotated probes with limited FC? Duplicates have not been removed and some show conflicting trends. While the statistical measures are appropriate from a general sense; adding an absolute FC criteria (1.2?) and additional cleaning of the data would be much appreciated and provide a better sense of mechanism.

While the plots in Figure 1 are interesting; I would still suggest the use of heatmaps on an individual level. These will provide a sense of variability and consistency in patterns. All data and separated by sex would provide value.

While the PCA clearly shows sex differences; I do not agree that the unsupervised PCA justifies combining 0.5 and 50 for their statistical analysis. I would suggest using other methods and then showing in a supervised manner how the two groups compare. Fig 1B indicates no reason that these two groups should be combined; besides the need to gain statistical power.

Line 331: Table 3. IPA analysis should include DE targets.

General comment:

Reduce language in the discussion and reduce speculation.

I would recommend a native-English speaker to review. There are several mistakes regarding tense and grammar.

Reviewer #4

(Remarks to the Author)

This article has been reviewed before by 3 reviewers who have provided detailed comments. The authors have tried to address the issues but have not made satisfactory changes. The study attempts to examine the effects of two doses of BPA exposure during pregnancy and lactation on bone marrow gene expression and other circulating biomarkers and parameters

in the offspring when they are a year old.

The study design is straightforward and there are enough number of dams in each group. Since the dam is the statistical unit, data from one male and one female pup from each dam could be used for the analyses. Tissues from a few pups of each sex from each dam could have been pooled for microarray analysis. However, the authors have chosen to pool data from the BPA 0.5 and BPA 50 groups. This is especially true for the figures. This is confusing the reader: Which dose is more dangerous, is it the BPA 0.5 or the BPA 50? Or are both doses equally dangerous? This is critical to establish because the clear feminization and masculinization of bone marrow gene expression as demonstrated by the authors may be potentially useful for making policy changes and recommendations. The authors have separated the BPA 0.5 and the BPA 50 groups in the supplementary tables, they need to do the same for the figures which are included in the manuscript.

Separating the data from two groups would require a different statistical analysis (2-way ANOVA). The authors need to use a statistical consultant.

The quality of all the figures is poor. Labeling of the X and Y axes needs to be bigger and clearer. Y axis titles need to be complete. For example, in figure 1 the Y axis is labelled as "intensity". The authors need to be more descriptive, for example: "Intensity of gene expression". While data from all control animals are included for figures 3-5, only a limited number of animals are included for the pooled BPA-treated groups. Heat maps need to be used for demonstrating differences in gene expression between the groups as the reviewers state. Probe values are not good indicators.

The manuscript requires considerable editing for syntax and grammar. It would be easier for the authors to use a language-editing service. An edited version of the introduction is attached for your convenience. The discussion needs to be condensed to 5 pages.

Version 2:

Reviewer comments:

Reviewer #3

(Remarks to the Author)

Thank you for addressing my comments point by point. The revised manuscript is much improved. However, I still have concerns regarding interpretation and presentation of the results.

Key comments.

Need more transparency regarding transcriptomic dose-response relationships. The authors fail to provide enough data/figures that demonstrate the relationship between low and high dose effects. Looking at the overlap of significant features provides a preliminary assessment; however PCA/heatmaps showing both concentrations together would provide greater support for proposed relationships between low and high dose and related sex-reversal changes. Unsupervised PCA results do not seem to support sex-reversal conclusions and suggest distinct changes in dose groups (males only), with substantially more changes in male (50ug); subsequent results do not reflect this (Table 1). Unsupervised PCA indicates limited global changes in female; yet more DEGs are identified in female vs. males. Could the authors produce heatmaps and PCA plots showing connection between low and high dose groups for subset of DE genes identified in both dose groups in male and female and the two sexes separately? This may show these relationships more clearly and the underlying variance across samples.

Overinterpretation of microarray data. Probes intensities are measures of relative expression, but are not equivalent to absolute abundance. Text, figure legends, and presentation of the data should reflect this. For example, I would suggest using z-score or adjusted values (ratio to average control) in Fig 1B-E to enable proper interpretation vs. probe intensities.

While I understand there may be a desire to combine dose groups for clinical chemistry assessments and possibly there is appropriate justification, the findings do not support this. For example, transcriptomic findings suggest drastic differences between dose groups in female. If significant findings cannot be achieved with single dose groups, this should at least be clearly stated.

When adding a 1.2 fold change cutoff to their DE analysis, the investigators found that this created a bias towards DEGs "with lower expression levels". Does this mean that most of your important genes are <1.2 fold change induction with a less than conservative cutoff. This should be stated. Are you concerned without multiple testing correction that you are including a high rate of false positives in your results? Would it make sense to be more conservative and remove low-expressed probes from your analysis (increase cutoff).

If a more comprehensive IPA analysis cannot be provided, would it be possible to provide additional outputs from KEGG (DAVID or ENRICH databases) that support your findings? It would be interesting to see the overlap in enriched categories (biological processes) identified in each of the DEG subsets and the overlap in significance.

Table 5. Could these be shown in a heatmap? What is the variability and fold changes associated with these genes?

Table 6. What are the genes linked to these pathways? How many up/down?

Reviewer #4

(Remarks to the Author)

The authors describe changes in metabolomic, lipidomic, bone marrow transcriptome and gene expression profiles in rats prenatally exposed to two low doses of BPA (0.5 and 50 µg/kg BW) and provide evidence to show that gene expression and lipid and hormonal profiles are altered by dose and sex. A lot of data is presented but presentation can be improved.

They state in the methods that they measured a variety of parameters such as BDNF, C-Peptide, Ghrelin, GLP1, PYY, Glucagon, Insulin, Leptin, IFN γ , IL1 β , IL4, IL5, IL6, IL10, IL13, CXCL1, TNF, CCL2, NGAL, TIMP1, and TSP1. Data is provided for TIMP1 and some cytokines have been described to be undetectable. Insulin levels are reported to be high in females and this should be included in figure 5.

The animal data seems to have been analyzed by two-way ANOVA to identify sex and treatment differences (like it should be), but this is not stated in the Statistical methods. The methods state that Student's t test was used to analyze all data other than human data, which is not true. Were statistical methods were used to compare the human and rat data to conclude that the metabolic profile is altered in the same way?

Table 1 should include comparisons in DEG between treated males and females and not just control males and females. Comparisons between males and females in 0.5 and 50 µg groups would be more meaningful and demonstrate sex differences in the presence of treatment.

What do the authors mean by "bone stiffness"? What parameter was used as an indication of this measure?

Lines 482-484. How do the authors conclude that "the few more changes induced by the lower BPA dose, identified by two-way ANOVA analysis of rat blood metabolites, may reflect an earlier disease stage rather than an increased hazard"?

When there are 18 control dams, 12 dams in the 0.5 group and 15 in the 50 group, why are there only 5 females and 3 males in each group included in the analysis for table 2?

While stating differences in gene expression and other parameters between males and females, the authors should include p values so the reader is able to discern the degree of these changes. Just stating "upregulated" and "downregulated" is not sufficient.

It is not appropriate to call the differential expression of genes as "sex reversal" without a phenotypic change or a change in genetic sex or behavior. Just because the changes induced by BPA in one sex results in values closer to the controls of the opposite sex, it does not mean that a sex-reversal is taking place.

It would be better to include a table providing values for parameters in both sexes in the different treatment groups with appropriate p values. In the discussion, it is also important to discuss the physiological relevance of the differential expression. Why is prenatal exposure to BPA producing this change? How is it expected to affect the health of the offspring? How does it increase the risk for metabolic syndrome? How are the observed changes in the bone marrow related to metabolic syndrome risk?

The authors state that the changes seen in rats are comparable to MetS in a human study. What specific changes are the authors referring to? These should be compared in a table.

Table 3: Showing the number of significant changes is irrelevant. It is important to show which particular parameter is significantly different and the corresponding p value.

Table 4: It is not clear what the authors mean by Matching and opposite.

Fig 1. It is difficult to see what is depicted in the heat maps. Are the same genes being compared in the different treatment groups with the control? Why is the pattern in controls different between 0.5 and 50 BPA comparisons?

Minor:

Lines 32-33: please remove "similar to a 100 times higher dose"

Line 33: Change to "BPA exposure induced sex-specific changes in gene expression..."

Line 37-38: Change to "...BPA exposure can cause metabolic syndrome specifically in males possibly by affecting T cell activity in a sex-specific manner".

Line 85: Change to "Bone maintains skeletal mass..."

Line 98-99: remove "use a hypothesis-free and unsupervised approach..."

Rephrase lines 106-107.

Line 121: replace "one dam per cage" with "individually".

Line 122: replace "one by one" with "individually".

Line 143: Remove "During the daytime"

Lines 166-167: Delete the line that starts with Humerus bone isolation.

Line 171-173: Change to "Total RNA was extracted from samples and hybridization and scanning were performed simultaneously..."

Line 195: remove all before Abcam Inc.

Line 203: "Lipid analysis was completed using the 1290..."

Line 204: "Nygren et al- please insert year

Line 241: Instead of ref 33, please insert author and year.

Line 288-289: "Introducing a DEG fold change..." is not clear. Please rephrase.

In general, please use a software to correct syntax and improve flow.

Version 3:

Reviewer comments:

Reviewer #3

(Remarks to the Author)

Dear Authors,

Thank you for the opportunity to review your revised manuscript. The study addresses an important and timely topic, and the exposures examined are physiologically relevant. The findings have potential implications for human health and regulatory policy. I appreciate your thoughtful and thorough efforts in responding to the prior critiques.

That said, I continue to find the presentation and interpretation of the data lacking in clarity and cohesion. Despite substantial revisions, key concerns remain regarding the transcriptomic analysis and its integration with the metabolomic findings. In some areas, the conclusions appear to extend beyond what the data can directly support. Given the complexity of the transcriptomic dataset and the potential impact of the findings, I would strongly encourage you to seek additional input from bioinformatics experts to strengthen the analysis and ensure appropriate interpretation.

Below are several specific points for further consideration:

1. Probe intensities reflect relative expression, not absolute abundance. While the manuscript states that the data were normalized, the heatmaps displaying normalized probe intensities and the tables listing "average expression" may misrepresent the data and could be misleading to readers.
2. The global PCA plots presented do not clearly support the conclusions regarding BPA-related effects. PCA plots focusing on differentially expressed gene subsets—especially those hypothesized to reflect sex-related shifts—would be informative.
3. The manuscript notes that most transcriptomic changes are <1.2-fold and that higher fold changes occur primarily among low-expressed probes. This raises concerns about the robustness of these findings and suggests that more conservative expression filters or fold-change thresholds may be warranted.
4. Heatmaps/Tables depicting relative differences between groups, rather than normalized intensity values, would improve the clarity and interpretability of the data.
5. Comparing different probes that map to the same gene may not be appropriate in the context of small fold changes. Consolidating multiple probes into a single representative value per gene would enhance clarity and analytical rigor.
6. Please clarify the total number of probes included on the array, how many were retained after filtering based on expression thresholds, and how many unique genes were ultimately represented in the analysis.
7. Including distribution plots of probe intensities before and after normalization would help demonstrate the effectiveness of preprocessing. Additionally, do sex-biased expression patterns persist when alternative normalization methods are applied? Were there initial differences in overall probe intensity distributions between groups that might confound interpretation?
8. How were the transcriptomic data annotated? What is the year and version?

Thank you again for your efforts. I hope these comments are useful as you continue to improve the manuscript.

Reviewer #4

(Remarks to the Author)

Thank you for highlighting the changes.

Minor changes are suggested:

Line 264: Two-way ANOVA was used to analyze rat plasma NMR and lipidomics data to identify sex and treatment differences. Student's t-test was used to analyze all other data

The revised manuscript looks good. Thank you for making the revisions.

Version 4:

Reviewer comments:

Reviewer #3

(Remarks to the Author)

Thank you for addressing my comments and concerns. Congratulations on your hard work!

I have one minor point. The term expression levels typically refers to absolute abundance. While I appreciate the modification to expression levels, this still may not accurately reflect the underlying data, which are based on probe intensities. I would suggest using probe intensities or relative expression to more precisely describe the values.

Reviewer #4

(Remarks to the Author)

The revised manuscript looks good and can be accepted for publication.

Author's response to reviewer's comments:

Reviewer #1 (Remarks to the Author):

Thanks to the editor for the invitation.

This study investigated sex-specific effects of BPA on bone marrow transcriptome and blood metabolome. However, the findings lack validation, the data results are not enough to support the conclusion, and the quality of the manuscript needs to be improved. Overall, this study is not enough to be published in a high-level journal.

The experimental design is puzzling. The entire study, including genetic testing and analysis, as well as the examination of metabolic data in the population, was taken from published articles and appears to be a recombination of previously published data.

The language of the manuscript needs to be improved.

Author reply: *The main findings are from the rat and are new data (microarray analysis, lipidomics, metabolomics, multiplex ELISA metabolomics, potassium, ALT and AST analysis). To further clarify the importance of the findings in relation to what is known we have add new paragraphs in the Introduction (line 54, 63, 71, 79), and Discussion (line 549, 585, 588, 599, 618, 624, 655).*

Reviewer #1

Abstract. The abstract section needs to reorganize the logic. And it is recommended that the abstracts do not cite references.

Author reply: *A clarification of the importance of this work is added to the Abstract (line 44). Reference citation has been removed from the abstract.*

Reviewer #1 Results. Both in the descriptive text of the results and in the graphical presentation, need to be further modified. The description of the results requires a summative description, streamlining, and highlighting the results. The presentation of the figures or tables suggests highlighting the point rather than listing all the results and leaving the reader looking for the resulting data.

Author reply: *Graphics for the experimental design have been added for increased overview and understanding (Fig. 1a). Improved plots of Fig. 1c,d,e,f and Fig. 2e,f are presented together with a better description of how to read these is added to the legends. Fig. 3 and 4 are now cleaned from less-significant data and presented as box-plots. Fig. 5 is cleaned from less-significant data and error-bars have been added. Together with new clarifying wordings in the Result text (line 270, 272, 295, 298, 300, 301, 306, 333, 339, 342, 386, 396, 437), for a better understanding of the logic behind the work. New data (Extended data Fig. 1 and 5) showing PCA plots and data from 5-week old siblings (see also Reviewer 3) to justify group pooling and highlight also the age effect after BPA exposure, has been added.*

Reviewer #1 Methods. The experimental methods of important experiments need to be detailed, for example, information on sample collection at the end of animal exposure, information on statistical analysis of data.

Author reply: *A paragraph describing the sample collection better has been added (line 143, and 166). A more detailed information regarding statistical analysis have also been added (line 241-249, 257, 264), see also below.*

Reviewer #2 (Remarks to the Author):

The authors use differential expression analysis and GSEA to provide evidence for significant effects from low-dose BPA exposure. Although this study includes important results on BPA exposure, there are several issues that have to be addressed before I can recommend the paper for publication. In particular, some statistical methods have to be described in more detail (see my comments below), and some figure and table captions contain insufficient information to fully understand what is displayed.

Major:

- Limma: the authors do not report if any confounding factors were included.

Author reply: *No confounding factors were included, this has been clarified (line 246).*

Reviewer #2 • Lines 901-902: No outlier effects were revealed by Quality Control (QC) plots -> which software / package / method did you use for the quality control?

Author reply: *We used the RMA-sketch version of the robust multi-array average (RMA) method first suggested by Li and Wong (Li C, Wong WH, Model-based analysis of oligonucleotide arrays: expression index computation and outlier detection., Proc Natl Acad Sci U S A. Jan 2; 98(1):31-6, 2001). This has been clarified (line 248).*

Reviewer #2 • Figure 3, 4, 5, Table S4, Table S5: are the p-values displayed adjusted for multiple hypothesis testing?

Author reply: *Neither the human, nor the rat results p-values were adjusted for multiple testing. The reason not to adjust for multiple testing in the small animal data set (exploratory part) is to potentially capture the biological parameters most influenced by BPA in these rats. This has been clarified (line 264). For the human data, we have now added (line 257) that if we apply an adjustment according to a false discovery rate (FDR) <0.05, the cut-off for significance for the nominal p-values given in the tables should be 0.012 for the MS-based metabolomics and 0.0086 for the NMR-based metabolomics. Thus, as you can see the majority of the relationships reported as significant are so also following this adjustment for multiple testing.*

Reviewer #2 • Supplementary Table S3: (adjusted) p-values are not reported.

Author reply: *These numbers are now added.*

Reviewer #2 Minor:

- Not common to add references in the abstract.

Author reply: *Reference citation has been removed from the abstract.*

Reviewer #2 • Line 130: change -> changed

- Line 201: patters -> patterns
- Line 402: indicate -> indicates

Author reply: *Thank you for bringing this forward, these errors have been corrected.*

Reviewer #2 • Line 775-776: time-mated -> do you mean age-matched?

Author reply: *We mean that the 45 female rats obtained from Charles River had been mated at the same time. To clarify this we change the wording to: mated at the same time (line 115).*

Reviewer #2 • R/Bioconductor and limma: please mention which version you used.

Author reply: *Release 3.6, this information is now available (line 246).*

Reviewer #2 • The authors report adjusted p-values. Which procedure was used? Benjamini-Hochberg?

Author reply: *Yes, Benjamini-Hochberg, this information is now available (line 247).*

Reviewer #2 • Captions extended data Tables and Figures, Fig 4 a-d, Fig 5: please explain from which statistical method these results are.

Author reply: *All analyses relating metabolites to presence/absence of the metabolic syndrome in humans have used logistic regression analysis. Figures (rat data) Student's t-test. This information has now been added.*

Reviewer #3 (Remarks to the Author):

Thank you for the opportunity to review COMMSMED-23-0432-T by Dr. Lind et al. Expanding previous work which highlighted the sensitivity of bone to BPA exposure, the enclosed study demonstrates that BPA developmental exposures (relevant to human exposure) alters the BM transcriptome and blood metabolome. Interestingly, the results show that BPA afflicts bone profiles/metabolic profiles in a sex-specific manner which may have implications regarding BPA contributions to immune cell dysfunction and human metabolic syndrome. Overall, the study is well written and of high importance for toxicology and risk assessment. The results are compelling; however, I have significant reservations regarding the interpretation of the results due to the preliminary nature of the transcriptomic analysis and missing data that supports sex-dependent phenotypes associated with BPA developmental exposure observed at 52 weeks.

Major concerns.

1. Assessments at 52 weeks likely reflect phenotypic differences as an outcome of developmental exposure vs. perturbations due to BPA exposure. This point is sometimes lost. What are the effects of BPA at each of these concentrations at birth, PND22, etc. prior to these assessments? Were any cellular/ molecular assessments completed during exposure period to understand how effects of BPA manifest into differences observed at 52 weeks? Could the authors provide supplemental experimental data or at least a summary of expected differences at these earlier timepoints?

Author reply: *Description of the phenotype at PND35 are available in published work, refs 13, 26 and 27 in the manuscript and this has been better clarified in the text (line 79, 588). We have further*

presented new data from lipidomic analysis of blood from PND35 to show distinct metabolic differences noted between young and aged BPA exposed rats. Notably, young BPA exposed females and males show a phenotypic overlap, which is in contrast to their aged siblings, this supports that developmental BPA exposure have lifelong effects according age as well as sex, new figure (Extended Data Fig. 5) and explained in the text (line 396, 585).

Reviewer #3 Could the authors add a schematic which illustrates the experimental design, emphasizing examination of BM and metabolic profiles at 48 weeks after last exposure (perinatal) and highlights differences in profiles between the two testing doses.

Author reply: *Graphics for the experimental design have been added for increased overview and understanding (new Fig. 1a).*

Reviewer #3 The authors have provided a preliminary analysis of the transcriptomic data. I would encourage using statistical tests and methods to optimally examine sex-dependent effects in the context of BPA testing concentrations. Interpretation of probe intensities is misconstrued. Need to better justify separation of male/female or combining of BPA dose groups.

Specific Comments

Transcriptomic analysis: Overall, the analysis is preliminary and the authors do not provide rationale to support performed analyses or conclusions. I would suggest using different statistical approaches to examine sex-dependent differences in the context of BPA exposure. For example, would PCA plots using all samples suggest stratification of samples by sex (in exposed and controls), justifying analysis to look at sex-specific changes? Provided Table 1 and Figure 1 provide suggest differences across groups, but poor indication of overall trends.

Author reply: *We have added two PCA plots to show: 1) the clear separation between sexes and 2) that the separation between BPA doses within each sex are less clear, especially for female groups (new Extended Data Fig. 1a,b). These results show that sex-specific analysis are valid and that pooling of BPA doses within each sex is justified, which have been clarified in the text (line 306).*

Reviewer #3 The number of DETs is very large for many of the comparisons. Did you assess with more stringent criteria to see if your conclusions will still be the same. For example, it would be interesting to incorporate more-restrictive p-value and FC (fold change). Is the p-value <0.05 corrected for multiple testing?

Author reply: *The main aim of the present study is to apply as an unbiased approach as possible. Thus, starting the analysis by using minimal stringency regarding (for most genes, the unknown) importance of how large should an expression change be to make a significant impact, and cut off was set to an average signal level of 1.0 (one). This will result in a higher than usual number of DETs/DPIs, which are potentially important. Applying this strategy produced very clear and convincing results, indicating that our strategy is valid. At this point, our opinion is that new focused analyses using selected higher stringency would dilute and delay the present remarkable results and is for future studies, as upon publication these data would be available through NCBI's webpage and useful to a wider group of scientists. The p-values have been adjusted using Benjamini-Hochberg (line 247).*

Reviewer #3 Data could be cleaned up. Analysis still includes probes IDs not linked to DET (see Table

2), potentially non-informative information. Were lowly expressed transcripts removed from the analysis? Why were analyses completed on the gene level? Or, can the authors provide information about unique genes vs. transcripts.

Author reply: *We have removed some less-significant results from Fig. 4 and 5, and moved some to a new Extended Data Fig. 4, to clarify the message. Also, Extended Data Fig. 2 and 3 have been stripped from less-significant results and combined to a new Extended Data Fig. 3. We added new information to Table 2. We believe that the most straightforward way to present/interpret changes in probe signal intensity/expression levels is to link it to the closest known gene. An average signal intensity below 1.0 (one) was used as a cut off, this has been clarified (line 241).*

Reviewer #3 Generalized linear models could be applied to assess dose-dependent changes (in both/either males and females) to identify DETs, while enabling robust power (using all data). More complex GLMs could be used to identify sex-specific changes that are differentially expressed between sexes in the context of BPA exposure (interactions).

Author reply: *As discussed above: Although very interesting and relevant, we believe that additional more complex analysis is for future studies.*

Reviewer #3 I find it very confusing that later on in the analysis (Figure 2) that the authors combine DETs from 0.5 and 50ug/kg groups. How is this justified based on Table 1/Figure 1 data? There seems to be very limited overlap in these profiles. Did BPA genes (0.5 vs. 50) change in a similar manner with BPA in males or females? Current justification (line 138) is due to large differences in sex vs. exposure (which is expected), and “the high overall similarity in DET regulation”. As for the latter point, where is this shown? Figure 1 suggests limited overlap (VENN; 5-10% overlap in males or females).

Overall, could you provide PCA plots and heatmaps that highlight differences/similarities among groups? Table 1 and Figure 2/3 provides number and comparisons, but difficult to see overall relationships.

Author reply: *As mentioned above: We have added PCA plots to show that although the separation between sexes is very clear, the separation between BPA doses within each sex is less clear, especially for female groups (new Extended Data Fig. 1a,b), indicating that pooling of data is justified. Our Fig. 1c-f and Fig. 2e-f are essentially “heatmaps” showing how BPA exposure (black dots) affect intensity/expression level compared to control females (red dots) and males (blue dots). These plots better and more clearly visualize masculinization and feminization, in our opinion. This has been clarified in the text (line 295-302) and in figure legends. “the high overall similarity in DET regulation” refers to the overall similarity of dot-distribution in Fig. 1c-f, this has been clarified (line 309).*

Reviewer #3 3. Justification of metabolic analysis. In its current format, the authors do not provide interpretation of the data; more of a data dump. Can the authors categorize results to provide a clear picture in how early BPA developmental exposures alter metabolic/lipidomic signatures later on in life?

Author reply: *We have more clearly described the reasons for presented results (line 386). Fig. 3 and 4 are now cleaned from less-significant data and presented as box-plots (some data is found in a new Extended Data Fig. 4). Fig. 5 is cleaned from less-significant data and error-bars have been added. Extended Data Fig. 2 and 3 have been cleaned and merged to a new Extended Data Fig. 3. To further*

clarify the importance of the findings in relation to what is known we have add new paragraphs in the Introduction (line 54, 63, 71, 79), and Discussion (line 549, 585, 588, 599, 618, 624, 655).

Reviewer #3 4. Increased description of quantitative changes is needed. The authors appropriately emphasize significance using p-values, however, could provide more quantitative data regarding fold change; change in abundances to understand biological relevance of altered endpoints.

Author reply: *The importance of FC (fold change) for biological relevance is not well understood for most genes and to minimize the misinterpretation of this unknown, we prefer at this stage not to speculate about this too much. Although we agree that, there are many interesting sub analysis that can be done with the present data, our opinion is that this is for future studies to not dilute the present overview results.*

Reviewer #3 Introduction

Line 61. Please define “huge”.

Author reply: *A more detailed description of the CLARITY-BPA program is now provided (line 63).*

Reviewer #3 Introduction

Can you note diversity and number of studies in meta-analysis?

Author reply: *We added the following sentence (with a reference) to the introduction: “A recent overview of published meta-analyses indicate that BPA exposure has seriously affected human health, most convincingly on preterm birth, obesity, allergy and kidney disease.” (line 54).*

Reviewer #3

Line 88. Is there evidence of specific mechanisms demonstrating how altered endocrine/nuclear receptor activity can lead to changes in bone?

Author reply: *BPA is a weak estrogen receptor agonist and estrogen is the major regulator of bone metabolism in both men and women (Khosla, S., Oursler, M. J. & Monroe, D. G. Estrogen and the skeleton. Trends Endocrinol Metab. 23, 576-81 2012). This is clarified (line 71).*

Reviewer #3 Line 269. Heatmaps which show changing/revert expression changes would greatly help support this point.

Line 273. Heatmap would greatly support this.

Author reply: *As mentioned above, our Fig. 2e,f are essentially “heatmaps” showing how BPA exposure (black dots) affect intensity/expression level compared to control females (red dots) and males (blue dots). These plots better and more clearly visualize masculinization and feminization, in our opinion. This has been clarified in the figure legend.*

As mentioned above: the importance of FC (fold change) for biological relevance is not well understood for most genes and to minimize the misinterpretation of this unknown, we prefer at this stage not to speculate about this too much, thus to keep it simple we have colored the gene names (red for downregulation and green for upregulation) in Table 4.

Reviewer #3 Line 898. What is a pm-gcpg method?

Author reply: *It is an IterPLIER background adjustment method, now more clearly described (line 243).*

Reviewer #3 Line 906. What is the cutoff criteria for DETs?

Author reply: *Only probes with a group average signal value above 1.0 (one) are analyzed (line 241).*

Reviewer #3 Tables/Figures

Table 1. What is the cutoff criteria for DETs? Any information regarding fold differences?

Author reply: *Average signal value below 1.0 (line 241). Fold differences are available in Supplementary Tables.*

Reviewer #3 Table 2. In using microarray probes, I would hesitate calling these “expression levels”. Probes intensities represent relative expression. Would it be possible to log₂ adjust to control group (average or male). What is the significance of these DETs? Would make more sense to separate control data (into supplemental table) and focus on BPA effects. Using GLM and interaction terms would enable the ability to identify commonly altered genes and sex-specific.

Author reply: *As correctly pointed out, expression levels might be misleading, thus we have changed from DETs to: differential probe intensities DPIs throughout the manuscript. The main aim with Table 2 is to highlight that BPA exposure induces opposite regulation, particularly in transcripts that separates female and male controls, thus we strongly feel that keeping the control results is very important.*

Reviewer #3 Table 3. Why are dose groups combined? Was this a separate analysis using limma to look at dose-dependent effects? What are the range of p-values signifying? Where are all of the other categories?

Author reply: *As discussed above: The new PCA plot (Extended Data Fig. 1a,b) now show further justification for pooling the BPA doses within each sex, beyond similarities in the Venn diagram and of plots in Fig. 1 (within each sex). As described in the header and in the methods this is a downstream analysis using the Ingenuity Pathway Analysis (Qiagen.com). We chose to present the most significant categories for this study.*

Reviewer #3 Table 4. Could this be shown in a heatmap? Sense of FC and significance should be provided or described.

Author reply: *As mentioned above: the importance of FC (fold change) for biological relevance is not well understood for most genes and to minimize the misinterpretation of this unknown we prefer at this stage not to speculate about this too much, thus to keep it simple we have colored the gene names (red for downregulation and green for upregulation) in Table 4. Fold change is available in Supplementary Table S3.*

Reviewer #3 Table 5. What are the importance of these different groups? What are the range in values? What is significant?

Author reply: *The point of Table 5 is to visualize the substantial overlap between all the significant blood omic-changes from the rat study, with top ranked blood omic-changes in humans with metabolic syndrome, now more carefully explained (line 437). As these rats and humans are on completely different diets, there is no sense in presenting specific value ranges at this point, in our opinion. All changes presented show statistical significance.*

Reviewer #3 Figure 1. Why are titles b-e labeled masculinization or feminization? Panels b-e are supplemental, difficult to read and interpret. Smaller dots would be appreciated. Probe values should be adjusted and heatmaps could be used to reflect differences among groups. I would encourage using a method which enables statistical power to identify BPA effects across doses; unless PCA plots clearly show a divergence among dose groups. Probe values do not necessarily correspond to abundance/expression levels.

Author reply: *These data are only investigating sex-biased genes and are thus labeled masculinization when showing how BPA exposed female probes (black dots) creeps toward the male intensity (blue line) and the red dots indicate female controls, and vice versa for feminization of males. Smaller dots are now provided. To clarify how to interpret these graphs we added the following to the figure legend: BPA altered sex-biased DPIs vs average intensity, masculinization/ feminization visualized as black-dots (BPA exposed) creeps towards opposite sex levels (blue/red solid line).*

Reviewer #3 Figure 2. Why are does groups combined after showing differences in Fig. 1? Do heatmaps illustrate sex-expression reversion?

Author reply: *As also discussed above: The new PCA plot (Extended Data Fig. 1a,b) now show further justification for pooling the BPA doses within each sex, beyond similarities in the Venn diagram and of plots in Fig. 1 (within each sex). The masculinization/feminization plots are essentially “heatmaps”, showing changes in intensity levels.*

Reviewer #3 Figure 3 and 4. Why are these data combined? What is the reasoning for these diverse measurements? Box plots showing single values would be helpful to see data.

Author reply: *The measurements in Figure 3 are to support the validity of the Ingenuity pathway analysis results presented in Table 3, now clearly explained at line 333. These are now presented as boxplots. Similarly, in Figure 4, to backup findings from NMR metabolomics, we performed a multiplex ELISA of metabolic proteins, now more clearly explained line 386. The measurements are now presented as boxplots.*

Reviewer #3 Figure 4a-d. What are the units? Why are these measurements grouped together?

Author reply: *Figure 4 is now boxplots and has been cleaned up from less-significant results to more clearly show the major findings. Units are now present in the plots. These are group together as they are all from NMR analysis.*

Reviewer #3 Figure 5. PCA? Difficult to read. Is this supplemental?

Author reply: *Figure 5 highlights the selective and extensive impact of BPA on male blood lipids, while females are unaffected. We strongly prefer the presentation as it is, easy to deduce the size of each affected/unaffected lipid.*

1 **Author's response to reviewer's comments:**

2

3 **Reviewer #2** (Remarks to the Author):

4

5 The authors have addressed my comments sufficiently.

6

7

8 **Reviewer #3** (Remarks to the Author):

9 Thank you for providing additional details and clarifying the importance of the results. The
10 manuscript is much improved and represents tremendous work; however, there are still concerns
11 regarding flow, rigor and evidence supporting the conclusions.

12 Introduction.

13 While the modifications have added some clarity, the text is difficult to follow. I would suggest
14 rewriting and checking that each paragraph is clearly providing the context that you are trying to
15 convey to the reader. There are flow issues. Would it make sense to start with Paragraph 2 (line 86)
16 discuss importance of bone and lack of knowledge in toxicity studies? Moving to BPA (general,
17 paragraph 2) and then low-dose effects in bone (paragraph 3)? Then, lead into overview of study
18 (paragraph 4)? The current Paragraph 1 is very difficult to read. Key concepts should be broken into
19 separate paragraphs: 1) BPA (general) and low-dose effects; 2) sensitivity of bone and lack of
20 knowledge at low dose.

21 **Author reply:** *The introduction is now edited and clarified; much thanks to reviewer #4.*

22

23 It should be very clear that previous work and the model builds on previous published studies
24 (phenotype at PND35, refs 13, 26, 27) and this extends this research. Can this be integrated more
25 clearly into the last paragraph?

26 **Author reply:** *A section at the end of the introduction has been added to clearly highlight the
27 published work on this rat model leading up to this study (lines 105-107).*

28

29 Methods

30 Line 166: Please verify. Were steps in the process randomized (RNA isolation, scanning of
31 microarrays)? Were males and females processed at the same time?

32 **Author reply:** *The details of RNA isolation, processing, hybridization, and scanning have been clarified
33 (lines 166-167 and 171-173). All RNA was extracted on one occasion, and all samples were hybridized
34 and scanned simultaneously.*

35

36 Results

37 Line 275: While the authors have stated that this is preliminary, are there concerns that the
38 interpretation of this microarray data is largely based on probes not assigned to genes of known
39 function. Also, includes untranslated RNA (small, pseudo, miR) and many non-annotated probes with
40 limited FC? Duplicates have not been removed and some show conflicting trends. While the
41 statistical measures are appropriate from a general sense; adding an absolute FC criteria (1.2?) and
42 additional cleaning of the data would be much appreciated and provide a better sense of
43 mechanism.

44 **Author reply:** *The data has now been cleaned, and all probes are associated with a known gene for*
45 *today's work and genes with unknown functions for future work. Results from analysis with FC criteria*
46 *1.2 have been added as a new table and a new figure (Extended Data Table 1. and Extended Data Fig.*
47 *1.).*

48
49 While the plots in Figure 1 are interesting; I would still suggest the use of heatmaps on an individual
50 level. These will provide a sense of variability and consistency in patterns. All data and separated by
51 sex would provide value.

52 **Author reply:** *The plots in Figure 1 have been exchanged for heatmaps on an individual level. We*
53 *have focused on data comparing females and males separately, as sex-specific reporting is needed in*
54 *studies of health and disease (New Ref 41). This has been clarified in the text (lines 342-343).*

55
56 While the PCA clearly shows sex differences; I do not agree that the unsupervised PCA justifies
57 combining 0.5 and 50 for their statistical analysis. I would suggest using other methods and then
58 showing in a supervised manner how the two groups compare. Fig 1B indicates no reason that these
59 two groups should be combined; besides the need to gain statistical power.

60 **Author reply:** *New analysis has been added, using FC 1.2 supervision of microarray data and 2-way*
61 *ANOVA analysis of the blood lipidomic and metabolomic results. These results further establish a*
62 *clear sex separation and striking similarities in the matched regulation (up or down) within each sex*
63 *despite the dose difference. We also clarified that published phenotypes on these rats consistently*
64 *show sex-specific effects. The lower dose appears to have a weaker or similar phenotype as the higher*
65 *dose. In agreement with this, a recent robust study from the CLARITY-BPA consortium used*
66 *quantitative unsupervised analysis to show that BPA exposure up to 25-250 µg/kg BW/d has the*
67 *same effect (Ref 39), which includes both doses used in this study. In addition, mixture-based*
68 *approaches have been suggested as an important way forward in endocrine-disrupting chemical*
69 *studies with a public health aim (New Ref 40). Based on these facts, analyzing a low-dose mixed*
70 *exposure group would, in our opinion, filter out more usable results as most human exposure is low-*
71 *dose and mixed and also increase the statistical power. The striking sex-specific results of*
72 *masculinization, feminization and lipidomics noted with the pooled sample, we believe, support the*
73 *correctness of these results. These thought processes have been clarified: Results line 334-345,*
74 *Discussion 478-495.*

75
76 Line 331: Table 3. IPA analysis should include DE targets.

77 **Author reply:** *Unfortunately, these data are no longer available, as the bioinformatician performing*
78 *that analysis is no longer available. In our opinion, this information would not change the overall*
79 *message of the manuscript, as these lists keep evolving and thus change over time.*

80
81 General comment:
82 Reduce language in the discussion and reduce speculation.
83 I would recommend a native-English speaker to review. There are several mistakes regarding tense
84 and grammar.

85
86 **Author reply:** *The discussion is now halved, edited, and clarified.*

87

88

89 Reviewer #4 (Remarks to the Author):

90

91 This article has been reviewed before by 3 reviewers who have provided detailed comments. The
92 authors have tried to address the issues but have not made satisfactory changes. The study attempts
93 to examine the effects of two doses of BPA exposure during pregnancy and lactation on bone
94 marrow gene expression and other circulating biomarkers and parameters in the offspring when they
95 are a year old.

96

97 The study design is straightforward and there are enough number of dams in each group. Since the
98 dam is the statistical unit, data from one male and one female pup from each dam could be used for
99 the analyses. Tissues from a few pups of each sex from each dam could have been pooled for
100 microarray analysis. However, the authors have chosen to pool data from the BPA 0.5 and BPA 50
101 groups. This is especially true for the figures. This is confusing the reader: Which dose is more
102 dangerous, is it the BPA 0.5 or the BPA 50? Or are both doses equally dangerous? This is critical to
103 establish because the clear feminization and masculinization of bone marrow gene expression as
104 demonstrated by the authors may be potentially useful for making policy changes and
105 recommendations. The authors have separated the BPA 0.5 and the BPA 50 groups in the
106 supplementary tables, they need to do the same for the figures which are included in the
107 manuscript.

108 Separating the data from two groups would require a different statistical analysis (2-way ANOVA).
109 The authors need to use a statistical consultant.

110 **Author reply:** *We have added results from a 2-way ANOVA analysis of blood lipidomics and*
111 *metabolomics. To identify a low-dose effect more clearly, we analyzed each BPA dose separately,*
112 *comparing it to controls. Results are in the new Table 3 and show that the lower dose (BPA0.5)*
113 *induces a few more changes than the higher dose (BPA50). Results also show fewer associations to*
114 *sex in the analysis of the higher dose, which might indicate that this dose has already diminished the*
115 *sex differences. This, together with published phenotypes on these rats and other studies on BPA-*
116 *exposed rats, agrees with that of the two doses used here; stronger phenotypes have been noticed*
117 *with the higher dose or similar effects with both doses, albeit always different between sexes. See*
118 *also the reply to Reviewer #3 above (line 60, this document). This has been clarified in the text:*
119 **Results line 334-345, Discussion 478-495.**

120

121 The quality of all the figures is poor. Labeling of the X and Y axes needs to be bigger and clearer. Y
122 axis titles need to be complete. For example, in figure 1 the Y axis is labelled as "intensity". The
123 authors need to be more descriptive, for example: "Intensity of gene expression".

124 **Author reply:** *The quality of the figures and the labels for the Y-axis have been improved. New Figures*
125 *3 and 5 now also differentiate samples from BPA0.5 and BPA50 in the pooled group results as*
126 *different colors of the triangles.*

127

128 While data from all control animals are included for figures 3-5, only a limited number of animals are
129 included for the pooled BPA-treated groups.

130 **Author reply:** *Data for Figures 4 and 5 contains essentially results from all animals, except a few*
131 *without a measurable signal or extreme values. Data points in Figure 3 are fewer as the amount of*
132 *plasma from some animals was insufficient to be analyzed by multiplex ELISA. In addition, the*
133 *multiplex ELISA generated extreme values (which are not included in the analysis) more often, as*
134 *described in the document: Reporting Summary.*

135

136 Heat maps need to be used for demonstrating differences in gene expression between the groups as
137 the reviewers state.

138 **Author reply:** *We have added heatmaps in Figure 1.*

139

140 Probe values are not good indicators.

141 **Author reply:** *We have switched to differentially expressed genes (DEG).*

142

143 The manuscript requires considerable editing for syntax and grammar. It would be easier for the
144 authors to use a language-editing service. An edited version of the introduction is attached for your
145 convenience. The discussion needs to be condensed to 5 pages.

146 **Author reply:** *The discussion has been halved, edited, and clarified. We thank this reviewer for*
147 *supplying an edited version of the introduction.*

148

149

Author's response to reviewer's comments:

Reviewer #3 (Remarks to the Author):

Thank you for addressing my comments point by point. The revised manuscript is much improved. However, I still have concerns regarding interpretation and presentation of the results.

Key comments.

Need more transparency regarding transcriptomic dose-response relationships. The authors fail to provide enough data/figures that demonstrate the relationship between low and high dose effects. Looking at the overlap of significant features provides a preliminary assessment; however PCA/heatmaps showing both concentrations together would provide greater support for proposed relationships between low and high dose and related sex-reversal changes. Unsupervised PCA results do not seem to support sex-reversal conclusions and suggest distinct changes in dose groups (males only), with substantially more changes in male (50ug); subsequent results do not reflect this (Table 1). Unsupervised PCA indicates limited global changes in female; yet more DEGs are identified in female vs. males. Could the authors produce heatmaps and PCA plots showing connection between low and high dose groups for subset of DE genes identified in both dose groups in male and female and the two sexes separately? This may show these relationships more clearly and the underlying variance across samples.

Author's reply: *In Figure 1, new heatmaps showing both concentrations side by side reveal BPA-associated patterns common to both treatments compared to the controls (Fig. 1b,c). In addition, the hierarchical clustering of the pooled groups shows that both doses are mixed, particularly in the female group (Fig. 1d,e).*

Furthermore, Table 1 has been expanded to include additional comparisons, facilitating a better understanding of group overlaps and sex effects. This table shows that the difference in up- and downregulation is highest in control males compared to control females, indicating that BPA exposure reduces transcriptomic sex differences in the bone marrow. Looking at All DEGs (see Table 1), the most apparent pattern is the strong trend of the reduced number of DEGs differences comparing exposed males (or females) with opposite-sex controls, showing that the higher dose (50ug) is more similar to opposite-sex controls compared to the lower dose (0.5ug), with a 27-67% reduction of DEG numbers. In addition, there is a lesser difference between the 50ug dose groups than the 0.5ug dose groups (64% DEG number reduction). These results suggest trends of female masculinization and male feminization caused by BPA exposure. It also suggests that the higher dose is more effective than the lower dose.

Focusing on only Sex-biased DEGs (Table 1), the trends are the same. However, it now shows that both BPA doses in each sex are distinctly more similar to each other than any other group (at least 68-71% fewer DEGs compared to either control), suggesting that both doses of BPA exposure have similar effects, particularly regarding female masculinization and male feminization.

*This has been clarified in Results: **lines 281-296**.*

Reviewer #3 Overinterpretation of microarray data. Probes intensities are measures of relative expression, but are not equivalent to absolute abundance. Text, figure legends, and presentation of the data should reflect this. For example, I would suggest using z-score or adjusted values (ratio to average control) in Fig 1B-E to enable proper interpretation vs. probe intensities.

Author's reply: *The raw expression values are normalized. To clarify this, we have rephrased the relevant section in the Methods section, **lines 242-243**.*

Reviewer #3 While I understand there may be a desire to combine dose groups for clinical chemistry

assessments and possibly there is appropriate justification, the findings do not support this. For example, transcriptomic findings suggest drastic differences between dose groups in female. If significant findings cannot be achieved with single dose groups, this should at least be clearly stated.

Author's reply: *New results (Figs. 1 b,c,d,e) from heatmaps and hierarchical clustering reveal overlap between BPA doses within each sex.*

The main focus of this manuscript is to explore the sex-biased effects of BPA. New results in Table 1 indicate that BPA dose overlap within each sex is particularly high for sex-biased genes, and that the higher dose (BPA50) has a more potent effect.

New results from DEG analysis in the ENRICH database, utilizing the readouts of Wikipathways 2024 human, KEGG 2021 human, and CellMarker 2024. This analysis reveals a distinct functional overlap between male groups (see the new Supplemental Table S2). The overlap within exposed female groups is present but not as clear. It further suggests a T cell phenotype in both sexes.

To clarify our choice of pooling of samples, the sex-specific focus and a more clinically relevant approach, over dose-specific effects, we added the following paragraph to the text: lines 364-370 "So far, results indicate consistent (opposite) sex-specific effects from BPA-exposure, together with similarities in gene regulation within BPA-doses in the same sex, particularly regarding sex-biased gene expression. Notably, the phenotypic changes observed in these rats are also consistently sex-specific and more pronounced with the BPA50 dose^{14,27,29}. Thus, bone stiffness (the ability of a bone to resist bending in a 3-point bending test) and bone marrow fibrosis are the most robust phenotypes observed in female rats, and with both doses¹⁴."

And lines 374-377: "Here, we chose to explore further the sex-biased similarity between the BPA0.5 and BPA50 doses, increase statistical power, and mimic human exposure by analyzing mixed exposure groups (more relevant for humans) instead of focusing solely on the specific dose effects observed."

Reviewer #3 When adding a 1.2 fold change cutoff to their DE analysis, the investigators found that this created a bias towards DEGs "with lower expression levels". Does this mean that most of your important genes are <1.2 fold change induction with a less than conservative cutoff. This should be stated. Are you concerned without multiple testing correction that you are including a high rate of false positives in your results? Would it make sense to be more conservative and remove low-expressed probes from your analysis (increase cutoff).

Author's reply: *It has been made clear that the most significant differences measured are less than a 1.2-fold change, lines 304-305. Using these DEGs, we demonstrate that more than 99% of comparable DEGs are regulated towards levels characteristic of the opposite sex in both sexes. These numbers alone make us confident that the rate of false positives is low. This is explained in the text: lines 415-417.*

Reviewer #3 If a more comprehensive IPA analysis cannot be provided, would it be possible to provide additional outputs from KEGG (DAVID or ENRICH databases) that support your findings? It would be interesting to see the overlap in enriched categories (biological processes) identified in each of the DEG subsets and the overlap in significance.

Author's reply: *New results from DEG analysis in the ENRICH database, utilizing the readouts of Wikipathways 2024 human, KEGG 2021 human, and CellMarker 2024. This analysis reveals a distinct functional overlap between male groups (see the new Supplemental Table S2). The overlap within exposed female groups is present but not as clear. It further suggests a T cell phenotype in both sexes.*

Reviewer #3 Table 5. Could these be shown in a heatmap? What is the variability and fold changes associated with these genes?

Author's reply: *The old Table 5, now named Table 6, is presented as a heatmap and includes DEG expression levels and p-values.*

Reviewer #3 Table 6. What are the genes linked to these pathways? How many up/down?

Author's reply: *The answer to this question in the previous revision was: Unfortunately, these data are no longer available, as the bioinformatician performing that analysis is no longer available. In our opinion, this information would not change the overall message of the manuscript, as these lists keep evolving and thus change over time.*

Instead, we present new results from the ENRICH analysis of individual and pooled groups in a new Supplemental Table S2, which lists the genes associated with the pathways mentioned.

Reviewer #4 (Remarks to the Author):

The authors describe changes in metabolomic, lipidomic, bone marrow transcriptome and gene expression profiles in rats prenatally exposed to two low doses of BPA (0.5 and 50µg/kg BW) and provide evidence to show that gene expression and lipid and hormonal profiles are altered by dose and sex. A lot of data is presented but presentation can be improved.

They state in the methods that they measured a variety of parameters such as BDNF, C-Peptide, Ghrelin, GLP1, PYY, Glucagon, Insulin, Leptin, IFN γ , IL1 β , IL4, IL5, IL6, IL10, IL13, CXCL1, TNF, CCL2, NGAL, TIMP1, and TSP1. Data is provided for TIMP1 and some cytokines have been described to be undetectable. Insulin levels are reported to be high in females and this should be included in figure 5.

Author's reply: *Insulin results are now included in Figure 5.*

Reviewer #4 The animal data seems to have been analyzed by two-way ANOVA to identify sex and treatment differences (like it should be), but this is not stated in the Statistical methods. The methods state that Student's t test was used to analyze all data other than human data, which is not true. Were statistical methods used to compare the human and rat data to conclude that the metabolic profile is altered in the same way?

Author's reply: *The missing information about the use of two-way ANOVA has been corrected, lines 264-265. No statistical method was used to compile Table 7, which is based on separate results in humans and rats.*

Reviewer #4 Table 1 should include comparisons in DEG between treated males and females and not just control males and females. Comparisons between males and females in 0.5 and 50 µg groups would be more meaningful and demonstrate sex differences in the presence of treatment.

Author's reply: *Table 1 has been expanded to include additional comparisons, facilitating a better understanding of group overlaps and sex effects. This table shows that the difference in up- and downregulation is highest in control males compared to control females, indicating that BPA exposure reduces transcriptomic sex differences in the bone marrow. Looking at All DEGs (Table 1, top), the*

most apparent pattern noticed is the strong trend of the reduced number of DEGs differences comparing exposed males (or females) with opposite-sex controls, showing that the higher dose (50ug) is more similar to opposite-sex controls compared to the lower dose (0.5ug), with a 27-67% reduction of DEG numbers. In addition, there is a lesser difference between the 50ug dose groups than the 0.5ug dose groups (64% DEG number reduction). These results suggest trends of female masculinization and male feminization caused by BPA exposure. It also suggests that the higher dose is more effective than the lower dose.

Focusing on only Sex-biased DEGs (Table 1, bottom), the trends are the same. However, it now shows that both BPA doses in each sex are distinctly more similar to each other than any other group (at least 68-71% fewer DEGs compared to either control), suggesting that both doses of BPA exposure have similar effects, particularly regarding female masculinization and male feminization.

This has been clarified in Results: lines 281-296.

Reviewer #4 What do the authors mean by “bone stiffness”? What parameter was used as an indication of this measure?

Author's reply: Bone stiffness is measured in Newtons per millimeter (N/mm) using a 3-point bending test, which indicates the force required to bend a bone a specific distance. To clarify this, the following section has been added to lines 364-365 in Results: “(the ability of a bone to resist bending from a 3-point bending test)”.

Reviewer #4 Lines 482-484. How do the authors conclude that “the few more changes induced by the lower BPA dose, identified by two-way ANOVA analysis of rat blood metabolites, may reflect an earlier disease stage rather than an increased hazard”?

Author's reply: This is based on the fewer and weaker phenotypes observed in the rats exposed to the lower BPA dose (refs 14, 27, 28, 29). To clarify this, we rephrased this section, which now reads: “The higher BPA dose (although considered low) exhibits a stronger phenotype^{14,27,28,29}. Thus, as the lower BPA dose exhibits fewer and weaker phenotypes, the more numerous plasma changes induced by the lower BPA dose, identified by two-way ANOVA analysis of rat blood metabolites, may reflect an earlier disease stage rather than an increased hazard.”, lines 524-528.

Reviewer #4 When there are 18 control dams, 12 dams in the 0.5 group and 15 in the 50 group, why are there only 5 females and 3 males in each group included in the analysis for table 2?

Author's reply: This Table is from microarray analysis conducted on 24 animals, comprising three groups of 5 females per group and three groups of 3 males per group. To clarify this, we added this information in the Methods section, lines 168-170.

Reviewer #4 While stating differences in gene expression and other parameters between males and females, the authors should include p values so the reader is able to discern the degree of these changes. Just stating “upregulated” and “downregulated” is not sufficient.

Author's reply: Gene expression levels and p-values are now included in Tables 3, 5, and 6, for easier interpretation of results. In addition, p-values are included in the text.

Reviewer #4 It is not appropriate to call the differential expression of genes as “sex reversal” without a phenotypic change or a change in genetic sex or behavior. Just because the changes induced by

BPA in one sex results in values closer to the controls of the opposite sex, it does not mean that a sex-reversal is taking place.

Author's reply: *The expression "sex reversal" has been removed and replaced with "female masculinization and male feminization."*

Reviewer #4 It would be better to include a table providing values for parameters in both sexes in the different treatment groups with appropriate p values.

Author's reply: *Gene expression levels and p-values are now included in a new Table 5, presented in a heatmap style for more straightforward interpretation of the results.*

Reviewer #4 In the discussion, it is also important to discuss the physiological relevance of the differential expression. Why is prenatal exposure to BPA producing this change? How is it expected to affect the health of the offspring? How does it increase the risk for metabolic syndrome? How are the observed changes in the bone marrow related to metabolic syndrome risk?

Author's reply: *To clarify this, we added the following paragraph at the end of the discussion: lines 613-624 "The literature on how low-dose BPA exposure worsens health suggests that BPA induces epigenetic changes during the sensitive fetal period, which can persist across generations. These changes appear to impact the function of many organs, and the present work provides further insight into the mechanism underlying the effect on the bone marrow transcriptome and blood metabolites. BPA exposure appears to disrupt the subtle sex-specific differences in the highly evolved and tightly controlled regulation of gene expression, which occurs over a prolonged period, disrupting the specific female and male metabolisms towards a MetS phenotype. Our results indicate that bone marrow T cell function is altered in opposite directions in males and females exposed to BPA. As bone marrow metabolism has been shown to reflect overall body metabolism and regulate immune homeostasis in peripheral tissues²², it suggests that disturbed bone marrow function may significantly contribute to the metabolic changes induced by BPA exposure."*

Reviewer #4 The authors state that the changes seen in rats are comparable to MetS in a human study. What specific changes are the authors referring to? These should be compared in a table.

Author's reply: *This comparison is presented in Table 8.*

Reviewer #4 Table 3: Showing the number of significant changes is irrelevant. It is important to show which particular parameter is significantly different and the corresponding p value.

Author's reply: *A revised Table 3 includes the names of metabolites affected, their corresponding p-values, and the significant difference.*

Reviewer #4 Table 4: It is not clear what the authors mean by Matching and opposite.

Author's reply: *This has been clarified in the legend of Table 4: "Matching shows the number of DEGs regulated in the same direction (up/down) compared to female controls over male controls (FC/MC), i.e., female regulation."*

Reviewer #4 Fig 1. It is difficult to see what is depicted in the heat maps. Are the same genes being compared in the different treatment groups with the control? Why is the pattern in controls different between 0.5 and 50 BPA comparisons?

Author's reply: *Each heatmap is unique and contains significant genes only within a particular group comparison. Therefore, the controls will differ in each comparison. It should be used to identify similar patterns within each treatment group.*

Reviewer #4 Minor:

Lines 32-33: please remove "similar to a 100 times higher dose"

Line 33: Change to "BPA exposure induced sex-specific changes in gene expression..."

Line 37-38: Change to "...BPA exposure can cause metabolic syndrome specifically in males possibly by affecting T cell activity in a sex-specific manner".

Line 85: Change to "Bone maintains skeletal mass..."

Line 98-99: remove : "use a hypothesis-free and unsupervised approach.."

Rephrase lines 106-107.

Line 121: replace "one dam per cage" with "individually".

Line 122: replace "one by one" with "individually".

Line 143: Remove "During the daytime"

Lines 166-167: Delete the line that starts with Humerus bone isolation.

Line 171-173: Change to "Total RNA was extracted from samples and hybridization and scanning were performed simultaneously..."

Line 195: remove all before Abcam Inc.

Line 203: "Lipid analysis was completed using the 1290..."

Line 204: "Nygren et al- please insert year

Line 241: Instead of ref 33, please insert author and year.

Line 288-289: "Introducing a DEG fold change..." is not clear. Please rephrase.

In general, please use a software to correct syntax and improve flow.

Author reply: *All the minor changes have been made, and the text has been run through a language correcting software.*

Reviewers' comments:

Reviewer #3 (Remarks to the Author):

Dear Authors,

Thank you for the opportunity to review your revised manuscript. The study addresses an important and timely topic, and the exposures examined are physiologically relevant. The findings have potential implications for human health and regulatory policy. I appreciate your thoughtful and thorough efforts in responding to the prior critiques.

That said, I continue to find the presentation and interpretation of the data lacking in clarity and cohesion. Despite substantial revisions, key concerns remain regarding the transcriptomic analysis and its integration with the metabolomic findings. In some areas, the conclusions appear to extend beyond what the data can directly support. Given the complexity of the transcriptomic dataset and the potential impact of the findings, I would strongly encourage you to seek additional input from bioinformatics experts to strengthen the analysis and ensure appropriate interpretation.

Below are several specific points for further consideration:

1. Probe intensities reflect relative expression, not absolute abundance. While the manuscript states that the data were normalized, the heatmaps displaying normalized probe intensities and the tables listing “average expression” may misrepresent the data and could be misleading to readers.

Answer to 1: To minimize confusion, we added “expression levels” and “Differentially expressed gene levels” in the text to describe the heatmaps and better match the Tables, Results, line 281 and 358.

2. The global PCA plots presented do not clearly support the conclusions regarding BPA-related effects. PCA plots focusing on differentially expressed gene subsets—especially those hypothesized to reflect sex-related shifts—would be informative.

Answer to 2: In our opinion, Table 1 covers all possible comparisons with solid numbers, and more PCA-plots would not contribute significantly to clarification, in our opinion.

3. The manuscript notes that most transcriptomic changes are <1.2-fold and that higher fold changes occur primarily among low-expressed probes. This raises concerns about the robustness of these findings and suggests that more conservative expression filters or fold-change thresholds may be warranted.

Answer to 3: By using an Affymetrix microarray system designed for high-resolution measurements and high reproducibility. We believe, like Vivek Subbiah described in a Perspective in Nature Medicine (<https://doi.org/10.1038/s41591-022-02160-z>), that innovative strategies are needed to progress in evidence-based medicine. Here, we have included the often-ignored low-to-moderate-fold-change results of gene expression. Our results reveal clear sex-specific findings between individual groups, which were reinforced by combining sex-specific groups, indicating reliable results and remarkably extensive transcriptome changes resulting in female masculinization and male feminization later in life, following developmental exposure to low-dose bisphenol A. These findings are potentially

important for human health and should be shared with the scientific community for further investigation, in our opinion.

4. Heatmaps/Tables depicting relative differences between groups, rather than normalized intensity values, would improve the clarity and interpretability of the data.

Answer to 4: In our opinion, Table 1 now covers all possible comparisons, and further heatmaps/tables would not contribute significantly to clarification at this point.

5. Comparing different probes that map to the same gene may not be appropriate in the context of small fold changes. Consolidating multiple probes into a single representative value per gene would enhance clarity and analytical rigor.

Answer to 5: This would be a lot of not-so-straightforward work, and in our opinion, is not necessarily associated with enhanced clarity. We believe, at this point, this is suitable for future work.

6. Please clarify the total number of probes included on the array, how many were retained after filtering based on expression thresholds, and how many unique genes were ultimately represented in the analysis.

Answer to 6: A total of 36685 probe IDs with a result were used in the present study. This has been clarified in the Methods section, line 182.

More than 99% of the results were retained after filtering based on expression thresholds. This was clarified in the Methods section, line 246.

Our focus was on all statistically significant probe ID results, which were manually checked in Ensembl. As manual probe annotation is very time-consuming, the total number of unique probe IDs was not verified this time, and it is not essential at this point in the study, in our opinion.

7. Including distribution plots of probe intensities before and after normalization would help demonstrate the effectiveness of preprocessing. Additionally, do sex-biased expression patterns persist when alternative normalization methods are applied? Were there initial differences in overall probe intensity distributions between groups that might confound interpretation?

Answer to 7: The distribution of raw probe intensities for each array was visually evaluated for outliers and other types of variation. No such unusual variation was observed before normalization. Alternative normalization methods provide very similar results. The differences do not have a bias and are not of a magnitude that could explain the results reported in this manuscript.

8. How were the transcriptomic data annotated? What is the year and version?

Answer to 8: The array probes were annotated with updated NetAffx information at the time the experiments were performed in 2018, and statistically significant probes were manually annotated using Ensembl (ensembl.org) in 2024. This has been clarified in the Methods section, lines 182-184.

Thank you again for your efforts. I hope these comments are useful as you continue to improve the

manuscript.

Reviewer #4 (Remarks to the Author):

Thank you for highlighting the changes.

Minor changes are suggested:

Line 264: Two-way ANOVA was used to analyze rat plasma NMR and lipidomics data to identify sex and treatment differences. Student's t-test was used to analyze all other data

Answer: Thank you for this suggestion for improvement. This sentence is now included in the manuscript.

The revised manuscript looks good. Thank you for making the revisions.

Reviewers' comments:

Reviewer #3 (Remarks to the Author):

Thank you for addressing my comments and concerns. Congratulations on your hard work!

I have one minor point. The term expression levels typically refers to absolute abundance. While I appreciate the modification to expression levels, this still may not accurately reflect the underlying data, which are based on probe intensities. I would suggest using probe intensities or relative expression to more precisely describe the values.

Answer: Relative expression is now consistently used in the text, lines: 308, 310, 339, 341, 342, 353, 389, 420, 425, 428, 431 and in Tables and Figures.

It is described in the Methods section as coming from probe intensity. Line 271.

Reviewer #4 (Remarks to the Author):

The revised manuscript looks good and can be accepted for publication.